# Tumor-stroma interactions differentially alter drug sensitivity based on the origin of stromal cells

Benjamin D Landry[1], Thomas Leete[1], Ryan Richards[1], Peter Cruz-Gordillo[1] ID, Hannah R Schwartz[1], Megan E Honeywell[1] ID, Gary Ren[1], Alyssa D Schwartz[2], Shelly R Peyton[2] & Michael J Lee[1,*] ID

## Abstract

Due to tumor heterogeneity, most believe that effective treatments should be tailored to the features of an individual tumor or tumor subclass. It is still unclear, however, what information should be considered for optimal disease stratification, and most prior work focuses on tumor genomics. Here, we focus on the tumor microenvironment. Using a large-scale coculture assay optimized to measure drug-induced cell death, we identify tumor–stroma interactions that modulate drug sensitivity. Our data show that the chemo-insensitivity typically associated with aggressive subtypes of breast cancer is not observed if these cells are grown in 2D or 3D monoculture, but is manifested when these cells are cocultured with stromal cells, such as fibroblasts. Furthermore, we find that fibroblasts influence drug responses in two distinct and divergent manners, associated with the tissue from which the fibroblasts were harvested. These divergent phenotypes occur regardless of the drug tested and result from modulation of apoptotic priming within tumor cells. Our study highlights unexpected diversity in tumor–stroma interactions, and we reveal new principles that dictate how fibroblasts alter tumor drug responses.

**Keywords** drug sensitivity; precision medicine; triple-negative breast cancer; tumor microenvironment; tumor–stroma interaction

**Subject Categories** Cancer; Genome-Scale & Integrative Biology

**Mol Syst Biol. (2018) 14: e8322**

## Introduction

A central challenge in medicine is selecting which drug or drug combination will be the most beneficial for a given patient. In cancer therapy, this decision has typically been based on the anatomical origin of the disease, in combination with drug screening to empirically identify the most efficacious compounds. In most cases, drug response rates vary considerably, and the causes of this response variability remain unclear. Thus, for ongoing efforts to

improve precision/personalized medicine it is critical to identify features that contribute to the observed drug response variability.

Several studies now exist that have explored the relationship between tumor genetics or tumor gene expression and drug response (Lamb *et al*, 2006; Cohen *et al*, 2011; Barretina *et al*, 2012; Cancer Genome Atlas Network, 2012; Cancer Genome Atlas Network *et al*, 2012; Shah *et al*, 2012; Li *et al*, 2017). Many insights have been gained from these and other studies, but even collectively, these studies fail to create a clear understanding of the variable levels of sensitivity to commonly used chemotherapeutics (Innocenti *et al*, 2011; Jiang *et al*, 2016). An important consideration is that substantial non-genetic heterogeneity exists within tumors, and these influences are generally missed in studies that focus exclusively on tumor genomics. For instance, several classes of normal cells typically reside within tumors. It is increasingly recognized that many tumor phenotypes, including tumor initiation, epithelial-to-mesenchymal transition (EMT), metastatic potential, and drug sensitivity, are influenced by interactions between cancer cells and the normal cells residing within or near tumors (Kalluri & Zeisberg, 2006; Pallasch *et al*, 2014). Prior studies have typically highlighted the variable and unpredictable nature of tumor–stroma interactions, in which the drug sensitivity of cancer cells appears to depend on the particular combination of tumor cell, stromal cell, and drug used (Mcmillin *et al*, 2010). Thus, for efforts to improve precision medicine, a critical unmet need is to learn "rules" dictating how stromal cells influence the drug sensitivity of cancer cells.

Here, we develop a mixed coculture assay optimized to specifically quantify cell death, rather than cell proliferation, and we use this assay to characterize functional interactions between tumor cells, stromal cells, and anticancer chemotherapeutic agents. We report that stromal fibroblasts influence tumor drug response in two distinct and divergent manners: Some interactions result in drug resistance, while others cause drug sensitization. Importantly, these divergent influences were associated with the anatomical tissue from which the fibroblasts were harvested. Surprisingly, our data show that these distinct fibroblast-dependent phenotypes are conserved regardless of the identity or molecular target of the drug. These broad-spectrum changes to drug sensitivity result from modulation of mitochondrial apoptotic "priming", which changes the

---

1 Program in Systems Biology, Program in Molecular Medicine, Department of Molecular, Cell and Cancer Biology, University of Massachusetts Medical School, Worcester, MA, USA
2 Department of Chemical Engineering, University of Massachusetts, Amherst, MA, USA
*Corresponding author. Tel: +1 774 455 3870; E-mail: michael.lee@umassmed.edu

threshold for initiation of apoptosis in cancer cells. Taken together, our study highlights previously unappreciated principles, which dictate how stromal fibroblasts alter a tumor cell's drug response.

# Results

## Cell-intrinsic sensitivity to commonly used chemotherapy is similar for basal-like and mesenchymal-like TNBC cells

To explore cell non-autonomous regulation of drug sensitivity, we began by focusing on a tumor subclass that displays notable response heterogeneity, without clear mechanisms that underlie these differences. "Triple-negative" breast cancers (TNBCs) are the most chemosensitive subtype of breast cancer, but also the subtype with the shortest disease-free survival and lowest overall survival rates (Carey *et al*, 2007; Anders & Carey, 2008). This paradox is thought to result from heterogeneity within the TNBC subclass (Lehmann *et al*, 2011). Additionally, although TNBC can be further stratified into several definable groups which differ in chemosensitivity (Lehmann *et al*, 2011), it remains unclear which features are responsible for creating the variable drug sensitivity that is observed clinically.

To highlight this variability in the drug response, we selected a panel of ten TNBC cells from either the "basal-like" or "mesenchymal-like" expression classes (Perou *et al*, 2000; Heiser *et al*, 2009; Lehmann *et al*, 2011). Basal-like (BL) cells—sometimes referred to as "basal A", "basal-like 1", and "basal-like 2"—are defined by expression of basal or myoepithelial genes. These cells are highly proliferative, tend to have elevated expression of DNA damage response genes, and generally respond at higher rates to cytotoxic chemotherapies *in vivo* (Lehmann *et al*, 2011). Mesenchymal-like (ML) TNBCs—which includes "mesenchymal", "mesenchymal stemlike", and also "claudin-low" expression classes—are enriched for expression of genes related to EMT, and genes associated with stemness. These cells are more "aggressive" clinically, more de-differentiated, more metastatic, and more chemoresistant *in vivo* (Prat *et al*, 2010; Lehmann *et al*, 2011). Thus, we initially reasoned that identifying mechanisms which account for the variability in DNA damage sensitivity between the BL and ML subclasses may aid in patient stratification or help to identify new strategies for improving responses to these agents.

To identify features that contribute to differential DNA damage sensitivity between BL and ML cells, we began by profiling the response of TNBC cells to doxorubicin (also called Adriamycin), a topoisomerase II inhibitor that is commonly used in the treatment of TNBC. We suspected that if the observed clinical patterns of aggressiveness were due to intrinsic differences in drug sensitivity associated with these gene expression states, different levels of sensitivity to doxorubicin should be observed *in vitro*. Indeed, the least sensitive cells were HCC-1395, a TNBC of the ML expression state; the most sensitive cells were MDA-MB-468, a TNBC in the chemosensitive BL category (Fig 1A). In contrast, however, the rest of the cell lines tested were similarly sensitive to doxorubicin, regardless of their gene expression state. To see whether this was unique to doxorubicin, we also profiled responses to other topoisomerase inhibitors in this panel of cells. Overall, these data reveal

relatively similar levels of drug sensitivity across these 10 cell lines (Appendix Fig S1A). To more rigorously determine whether the patterns of sensitivity to these drugs could be used to distinguish BL versus ML cells, we performed hierarchical clustering using either the $EC_{50}$ or the maximum effect observed for each drug. This analysis also failed to correctly separate BL and ML cells based on their observed drug sensitivity profile (Fig 1B and Appendix Fig S1B).

The results from our *in vitro* analysis suggest that these BL and ML cells have similar sensitivity to commonly used chemotherapeutics. This, of course, is not in line with the expected observation that ML tumors respond at lower rates than BL tumors *in vivo* (Ahn *et al*, 2016). One explanation could be bias within our samples, as our dataset was comprised of a relatively small number of TNBCs. To address this, we also analyzed data publically available through the LINCS consortium, which include drug sensitivities for a larger panel of 24 BL or ML cell lines (11 BL and 13 ML) and a larger panel of common anticancer drugs (Fallahi-Sichani *et al*, 2013). These data also show that BL and ML cells have similar levels of sensitivity to topoisomerase inhibitors, specifically, or to all anticancer drugs, generally (Fig 1C and Appendix Fig S1D). Thus, taken together, these data highlight that the subtype-dependent differences in drug sensitivity, which may be expected given responses observed in patients, are generally not observed when these cells are grown in standard *in vitro* cell culture conditions.

Another potential explanation for the discrepancy between our data and the relative drug sensitivities that were expected could be that our cells were grown in 2D, rather than using 3D culturing conditions. It has generally been found that many cell behaviors differ when cells are grown in 2D versus 3D, and that 3D culture is in many ways a more accurate representation of the *in vivo* environment (Yamada & Cukierman, 2007; Fang & Eglen, 2017). To test whether growth in 3D recapitulates the expected distinction between BL and ML cells, we retested sensitivity to 10 topoisomerase inhibitors for TNBC cells grown as 3D colonies in a Matrigel growth environment. Growth of these TNBC cells in 3D colonies strongly altered drug sensitivity (Fig 1D). In some cases, a modest trend was observable in which ML cells appear less sensitive to drugs (e.g., camptothecin), but these trends were not statistically significant. The dominant trend was an overall desensitization to these drugs, without further refining the distinction between BL and ML cells ($P > 0.05$ for all drugs in 2D and 3D culture; Fig 1D and Appendix Fig S1C). This finding is consistent with prior studies, which demonstrated that growth in 3D induces a general resistance to drug-induced apoptosis (Weaver *et al*, 2002). Thus, taken together, our data highlight that these BL and ML TNBC cells are similarly sensitive to commonly used chemotherapeutics, at least when cultured in standard 2D or 3D monoculture conditions. This finding raises the possibility BL and ML subtype-specific responses to treatment are not cell-intrinsic properties, but rather a product of subtype-specific interactions between tumor cells and microenvironmental features.

## Coculture screen optimized to monitor cell death reveals widespread stromal influence on TNBC drug sensitivity

Based on the results of our *in vitro* drug screen of TNBC cells grown in monoculture, we aimed to test the hypothesis that differences

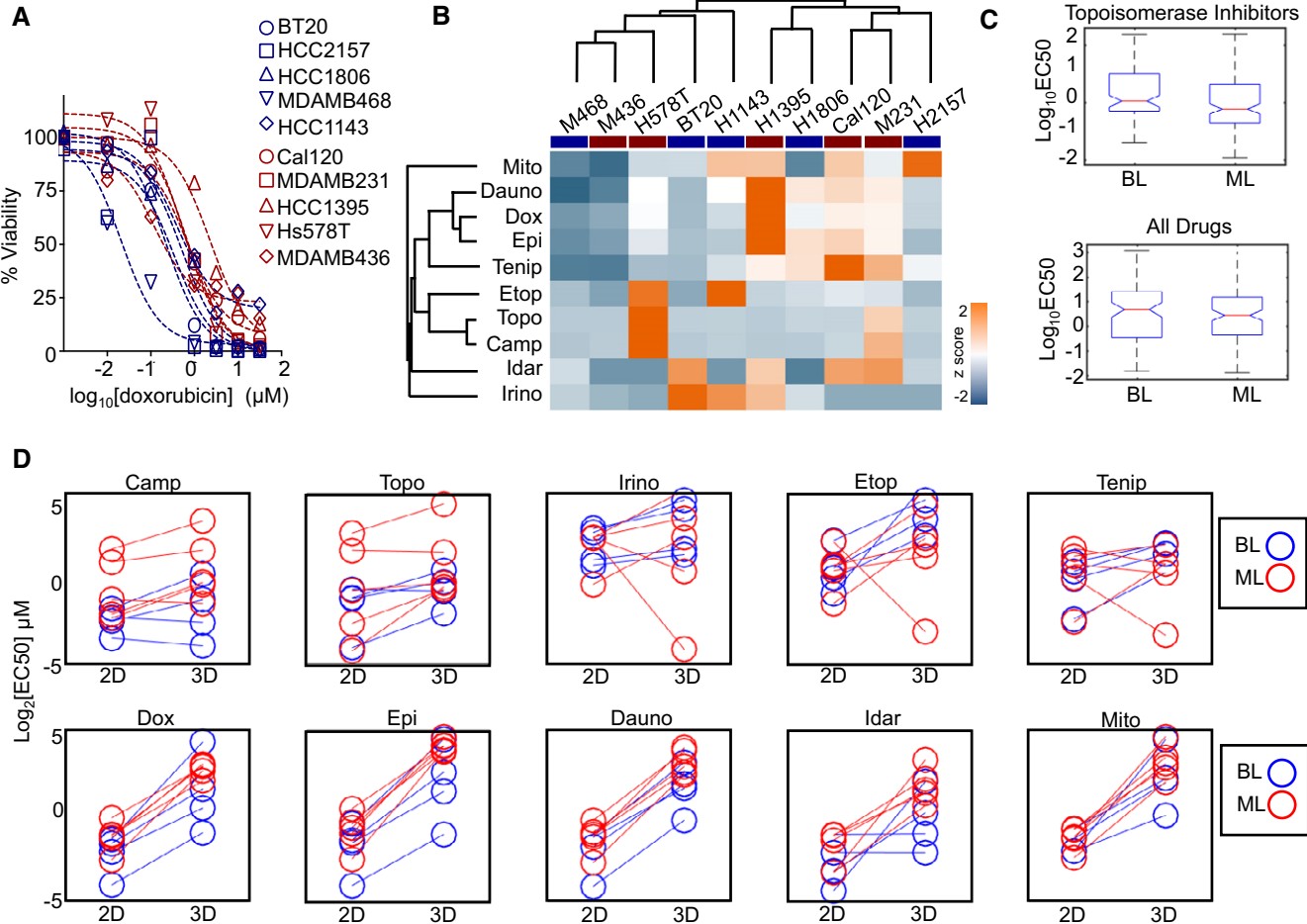

**Figure 1. TNBC cell sensitivity to topoisomerase inhibition is not well predicted by basal-like versus mesenchymal-like gene expression status.**

A  Relative sensitivity to doxorubicin. Panel of 10 TNBC cell lines from the basal-like (BL, blue) or mesenchymal-like (ML, red) gene expression subclasses were exposed to varying doses of doxorubicin. Cell viability quantified using CellTiter-Glo, 72 hours after drug exposure. Data are from biological triplicates.

B  Cell viability measured as in (A) for 10 common Topo I or II inhibitors. Data are z-scored $EC_{50}$ per drug. Dendrograms from hierarchical clustering shown for drugs and for cells (BL cells highlighted with blue bar; ML cells highlighted with red bar).

C  Sensitivity to topoisomerase inhibitors (top) or all drugs (bottom) in publicly available LINCS data. Data are representative of 24 TNBC cell lines (11 BL and 13 ML) and 67 total drugs (six Topo Inhibitors). In the boxplots, the red horizontal line represents the median; blue box show the $25^{th} - 75^{th}$ percentile range; whiskers show the full range of the data.

D  Drug sensitivity of TNBC cells grown in 2D or as 3D spheroids in soft Matrigel. Cell viability assessed using CellTiter-Glo, as in panel (A). For each drug shown, sensitivity is not significantly different between BL and ML cells, whether grown in 2D or 3D ($P > 0.05$ for all).

between the chemosensitivity of BL and ML cells are induced, in part, by cell non-autonomous influences. Several studies have suggested that interactions between tumor cells and components of the tumor microenvironment—including extracellular matrix, growth factors, and other stromal cell types—can alter sensitivity to chemotherapy (Weaver *et al*, 2002; Straussman *et al*, 2012; Nguyen *et al*, 2014). We focused on interactions between cancer cells and stromal fibroblasts, which are often the predominant stromal type found within the tumor microenvironment (Buchsbaum & Oh, 2016).

To identify tumor–stroma interactions that alter drug response, we initially used an *in vitro* coculture system that was successfully used to evaluate tumor–stroma–drug interactions (Straussman *et al*, 2012). In this experimental platform, cancer cells are genetically modified to express GFP, to facilitate rapid, quantitative, high-throughput, and cancer cell-specific measurement of drug response

dynamics. To pilot this study, we evaluated fibroblast influence on the response of BT-20 cells to targeted and cytotoxic chemotherapies. Our microscopy-based analysis revealed that sensitivity to both targeted and cytotoxic therapies was inhibited by coculture with HADF, a primary non-immortalized human fibroblast harvested from the adrenal gland (Appendix Fig S2A and B). Interestingly, analysis of these same cocultures using a fluorescence plate reader successfully captured only the fibroblast-dependent inhibition of erlotinib-induced growth arrest, but failed to capture the fibroblast-dependent inhibition of cytotoxic therapy (Appendix Fig S2C). Notably, in addition to missing the fibroblast influence in the context of chemotherapy, plate reader-based analysis also failed to capture the potent death that we observe following camptothecin exposure, with all measurements in the time course recording higher values than the initial pre-drug measurement (Appendix Fig S2C). The insensitivity of this screening approach was likely due to

the stability of GFP fluorescence, even after cell death (Appendix Fig S2D and E). These data indicate that measurements of GFP fluorescence using a fluorescence plate reader were not sufficiently sensitive for quantifying the degree of cell death in a population of cells.

Based on these results, we modified our coculture screen to optimize measurement of drug-induced cell death. We used JC-1, a dye that accumulates within mitochondria and is often used as a surrogate measure of apoptotic cell death (Fig 2A and B; Montero *et al*, 2015). At low concentrations, JC-1 exists as a monomer and yields green fluorescence; however, when accumulated at high concentrations within mitochondria, this dye forms aggregates, which yield red fluorescence. Thus, the red fluorescence of JC-1 reports cellular mitochondrial integrity, which is lost when cells activate apoptosis. To assess the suitability of JC-1 to quantify changes in the degree of cell death in coculture, we again piloted this assay on BT-20 cells treated with camptothecin in the presence or absence of HADF. Images of these cells taken prior to drug exposure confirm punctate

red fluorescence in BT-20, but not HADF, confirming that the dye is not exchanged between cells in coculture (Fig 2B). 96 h after exposure to camptothecin, the majority of BT-20 cells had significantly reduced JC-1 red fluorescence, suggesting that mitochondrial integrity has been compromised (Fig 2C). Importantly, JC-1 red fluorescence measured using a fluorescence plate reader was sufficiently sensitive for observing both the potent cell death of BT-20 cells in monoculture and the protective effect of HADF cells in coculture (Fig 2D).

To evaluate the role of stromal fibroblasts in DNA damage sensitivity, we selected six TNBC cell lines (three BL and three ML) that have relatively similar levels of sensitivity to DNA damage. These JC-1-labeled TNBC cells were grown in monoculture or in coculture with each of a panel of 16 primary human fibroblasts. Each culture was exposed to a four-point dose range of 42 anticancer drugs, which included at least one drug per class for all current FDA-approved breast cancer drugs (Tables EV1 and EV2). JC-1 red fluorescence was quantified at 8-h intervals for 72 h. In total, we

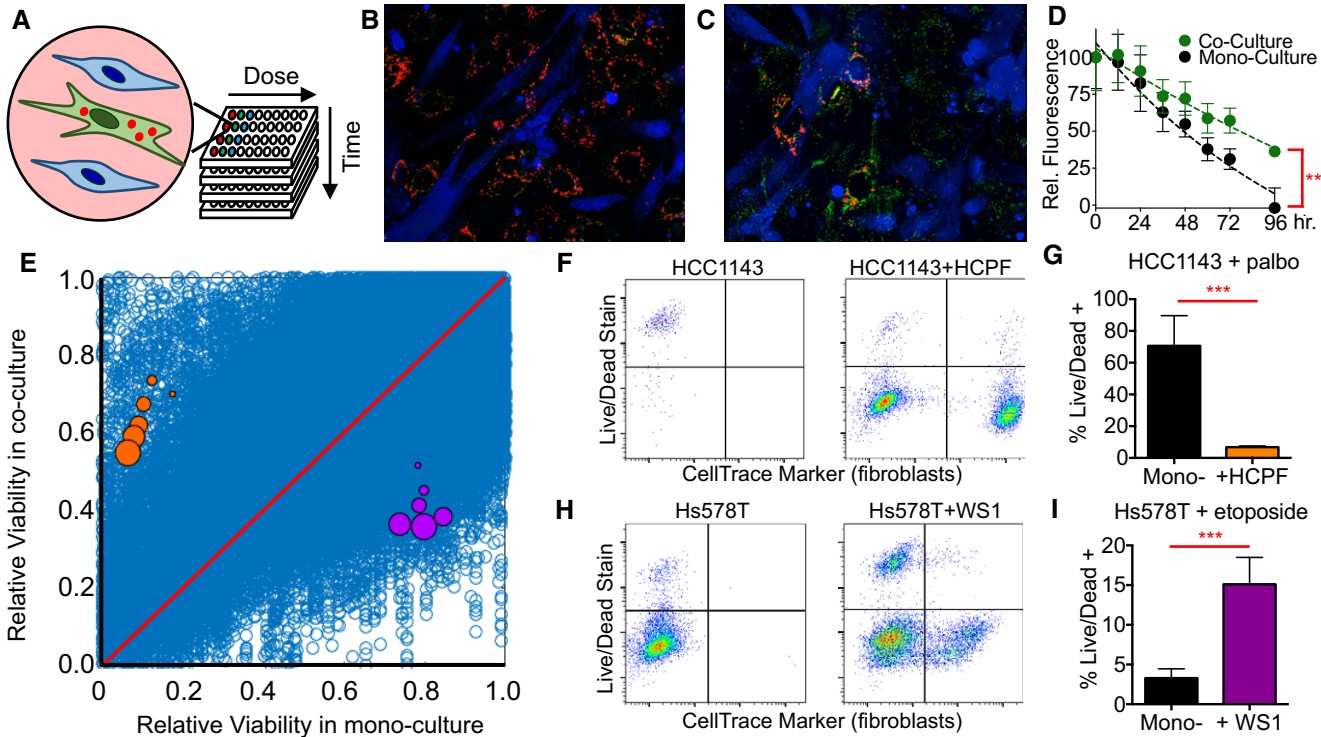

Figure 2.  Coculture screen to identify tumor–stroma interactions reveals divergent interactions between fibroblasts and TNBC cells.

A   Schematic of screen design. TNBC cell lines were labeled with JC-1, grown in monoculture or in coculture with primary fibroblast cells, and treated with one of 42 anticancer drugs. JC-1 fluorescence was monitored using a fluorescence plate reader at 8-hour intervals for 72 h.

B, C   Representative images of BT-20 cells (BL subtype) cocultured with HADF fibroblasts. BT-20 cells labeled with JC-1 dye; HADF labeled with a blue cell dye (CellTrace). Images taken before drug addition (B) or 96 h after exposure to 0.5 μM camptothecin (C).

D   Kinetic trace of JC-1 red fluorescence following exposure to camptothecin as in (B and C). Data are relative JC-1 red fluorescence, normalized to the well's average prior to drug addition. Data represent mean ± standard deviation for five biological replicates.

E   Total coculture screen data. Each blue dot represents a unique TNBC–fibroblast–drug measurement. 312,120 total measurements of drug response. Orange and purple dots highlight conditions validated in panels (F–I). Orange are HCC-1143 (BL) cells cocultured with HCPF and treated with palbociclib. Purple dots are Hs578T (ML) cocultured with WS1 and exposed to etoposide. For colored dots, increasing size represents longer drug exposure times.

F–I   Validation of coculture screening data. (F) Flow cytometry analysis of cell death using Live/Dead Blue stain. FarRed CellTrace was used to label and distinguish fibroblasts. (G) Quantification of percent TNBC cell death in experiment described in panel (F). (H and I) Flow cytometry and quantitative analysis, as in panels (F and G) for Hs578T +/− WS1 and treated with etoposide. Error bars represent standard deviation of biological replicates. *P*-value calculated using *t*-test. Data are from three biological replicates and error bars. ***P*-value < 0.05.

collected more than 300,000 measurements of drug sensitivity (Fig 2E, Appendix Fig S3A–C, and Table EV3). We found a strong overall correlation among biological replicates, indicating that the stromal influences observed were not due to measurement noise (Appendix Fig S3D). To identify TNBC–fibroblast interactions that significantly altered sensitivity, we used a statistical fold-change cutoff of 3× the standard deviation observed among replicates. This analysis identified 5,039 significantly changed drug responses (Appendix Fig S3E). As may be expected, this list of "hits" was significantly depleted for responses at early times (*i.e.,* 8 h), low doses (0.1 μM), and responses to anti-estrogen drugs (Tables EV3 and EV4). Non-response to anti-estrogen compounds is expected as TNBCs do not express estrogen or progesterone receptors.

The majority of TNBC cell–fibroblast interactions did not alter drug sensitivity (Appendix Fig S3A–C). Nonetheless, our screen revealed many striking phenotypes of strongly altered drug responses, and overall, our data cover nearly the entire landscape of possible positive and negative interactions (Fig 2E). To determine the reliability of these measurements, we selected a subset of these interactions to validate by flow cytometry. For example, our screen identified that palbociclib killed more than 80% of HCC-1143 cells, a basal-like TNBC, if applied to these cells in monoculture. However, this drug was rendered ineffective when HCC-1143 cells were cocultured with the fibroblast cell, HCPF, resulting in only a 20–40% decrease in cell viability (orange dots, Fig 2E). A flow cytometry-based analysis of cell death recapitulated this drug desensitization phenotype (Fig 2F and G). Additionally, our coculture screen identified instances in which the efficacy of etoposide is significantly improved in coculture conditions. For example, etoposide was ineffective in killing mesenchymal-like Hs578T cells in monoculture, but killed more than 50% of these cells grown in coculture with skin fibroblast cells, WS1 (purple dots, Fig 2E). This phenotype was interesting because prior studies have found that etoposide is minimally active in monoculture, which was surprising given the clinical utility of this compound (Lee *et al*, 2012). Flow cytometry-based analysis of cell death confirmed that etoposide-induced cell death in Hs578T is significantly enhanced by coculture with WS1 fibroblasts (Fig 2H and I). Thus, our coculture drug screen identified a spectrum of TNBC–fibroblast interactions that modulate the drug response of TNBC cells in both positive and negative directions (Appendix Figs S4–S6).

### Primary fibroblasts grown in coculture with TNBC cells display an activated phenotype and modulate drug sensitivity similar to cancer-associated fibroblasts (CAFs)

Our coculture drug screen was comprised mainly of normal primary fibroblasts. Cancer-associated fibroblasts (CAFs, also called "activated" fibroblasts, myoepithelial cells, or myofibroblasts) are thought to be largely distinct from primary fibroblasts in their ability to promote aspects of tumorigenesis and tumor progression (Kalluri & Zeisberg, 2006; Shiga *et al*, 2015). Thus, to address the relevance of our findings we also aimed to compare the behaviors of primary fibroblasts and CAFs, and in particular, the ability of these different cell types to modulate drug responses of TNBCs in coculture. We first asked whether our primary fibroblasts adopt an activated phenotype in culture. Prior studies have found that the unnatural environmental stiffness of *in vitro* culture can cause primary

fibroblasts to spontaneously adopt the activated phenotype (Huang *et al*, 2012). Using immunofluorescence microscopy, we determined α–smooth muscle actin (SMA) expression in fibroblasts, a marker of the activated fibroblast expression state that is commonly observed in CAFs. Robust but variable SMA positivity was observed for all primary fibroblasts tested (Appendix Fig S7A). To more precisely quantify percentages of SMA$^+$ fibroblasts, and to determine whether coculture modulated SMA positivity, we measured the percent of SMA$^+$ cells using flow cytometry (Appendix Fig S7B). The percentage of SMA$^+$ fibroblasts was relatively high, even when fibroblasts were grown in monoculture (median 63%; range 10–85%), and was consistently increased when fibroblasts were cocultured with TNBC cells (Appendix Fig S7C). Thus, primary fibroblasts grown in coculture with TNBC cells generally adopt the activated expression state that is commonly observed for CAFs.

To compare the degree to which primary or CAF cells modulate drug sensitivity of TNBC cells, we repeated a small portion of our coculture drug screen using one primary mammary fibroblast (HMF) and two breast cancer-associated fibroblasts (Hs343T and Hs578BST). Overall, the drug sensitization/desensitization profile revealed a similar pattern of relative drug sensitivity when compared to our initial coculture screen (Appendix Fig S7D). Additionally, primary and CAF cells modulated sensitivity to common anticancer drugs in a significantly correlated manner ($r = 0.46$; $P < 0.0001$). This suggests that, although primary fibroblasts and CAFs likely differ in many substantial ways, these different cell types were similar in the manner in which they alter drug sensitivity of associated cancer cells. This finding is consistent with a previous study by Polyak and colleagues, which found that patient-derived primary or CAF cells from breast or brain specimens similarly desensitized HER2-positive breast cancer cells to the HER2 inhibitor lapatinib (Marusyk *et al*, 2016).

### TNBC–fibroblast interactions are sufficient for inducing differential drug sensitivity between basal-like and mesenchymal-like TNBCs

Our profiling of TNBC cell lines grown in 2D or 3D monoculture failed to identify robust differences in the drug sensitivities between BL and ML cells (Fig 1). Since our coculture screen revealed strong positive and negative changes in the drug responses of these cells (Appendix Figs S4–S6), we asked whether these altered drug sensitivity profiles improved the resolution between BL and ML cells. To test this, we performed principal component analysis (PCA) on our coculture screening data. We reasoned that PCA would be beneficial due to the high dimensionality of our data (*i.e.,* multiple drugs, cells, coculture environments, doses, times). PCA uses the covariation structure of the data to reduce data dimensionality to a smaller number of "principal components", with each component being comprised of related information (Janes & Yaffe, 2006).

Principal component analysis of our coculture data reduced these complex observations to 10 principal components, with the first two components capturing 53% of the overall variation in drug sensitivity (Appendix Fig S8A). The projection of our data onto PC1 and PC2 revealed a clear separation of BL and ML cells (Fig 3A). Notably, this expected pattern was not visible in drug response data collected on these TNBC cells grown in monoculture (Figs 1B and 3B). Thus, our screening data reveal that coculture of TNBCs with

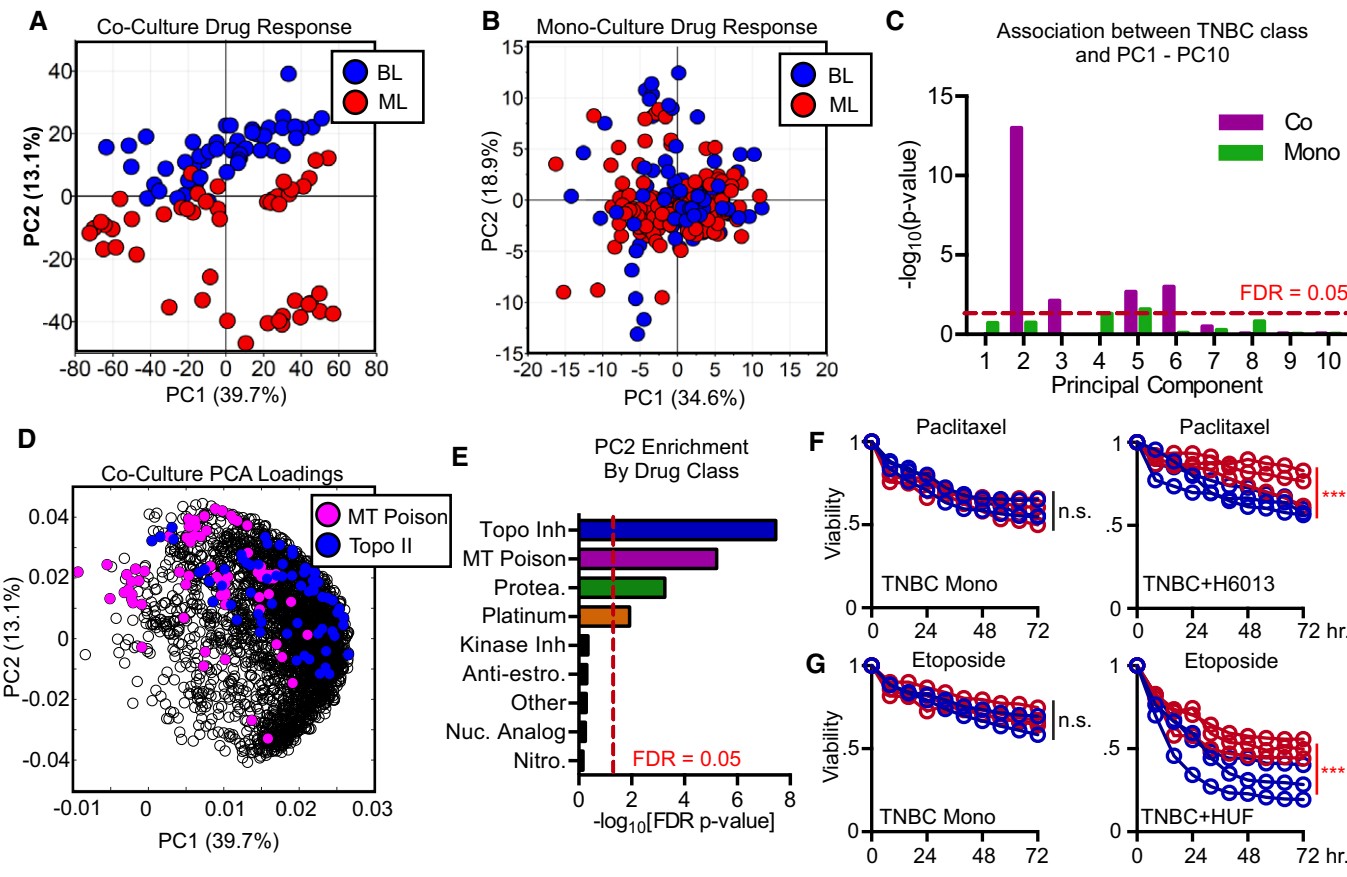

**Figure 3.  Drug responses of TNBC grown in coculture—but not in monoculture—are sufficient to distinguish basal-like and mesenchymal-like TNBC subclasses.**

A    PCA on coculture drug response. Projection of scores onto PC1 and PC2.

B    PCA on monoculture drug response.

C    Association between each principal component (PC1-10) and basal-like subclass.

D    Projection of PCA loading vectors from coculture drug response (panel A) onto PC1 and PC2. Microtubule poisons (MT Poison) and topoisomerase II inhibitors (Topo II) highlighted.

E    Association between PC2 and each drug class.

F, G   Examples of drug responses identified by PCA that differ between BL and ML cells. (F) Time course of TNBC viability following exposure to 10 μM paclitaxel in monoculture or coculture with H6013 (lung). (G) Time course of TNBC viability following exposure to 10 μM etoposide in monoculture or coculture with HUF (uterine).

Data information: In panels (C and E), enrichment *P*-values were calculated using Fisher's exact test. In (F and G), *P*-values were calculated using *t*-test. ***$P < 0.05$; n.s. = not significant.

fibroblast cells modulates drug sensitivity in a manner that enhances the distinction between BL and ML cells.

Because each principal component captures unrelated/orthogonal variation in the data, we also sought to determine which components were capturing variation associated with the BL/ML dichotomy. For example, the BL/ML distinction could be captured independently on all PCs. This would suggest that BL and ML cells respond differently to all drugs, in all coculture environments. Alternatively, it is also possible that the BL/ML distinction is restricted to one or a few PCs, suggesting that BL and ML cells respond in distinct ways only in the context of a subset of drugs and/or environmental conditions. Thus, to investigate this further, we quantified the degree of separation of BL/ML cells across all 10 PCs (Fig 3C). A strong association was identified for PC2, which captured approximately 13% of the data. Modest but significant associations with BL/ML were also found for PC3, 5, and 6, which each of which captured relatively small amounts of the dataset (Appendix Fig S8B). Thus, we focused on finding features in our data that were driving the association between PC2 and BL/ML cells. To achieve this, we inspected the PCA vector loadings, which report the degree to which each variable contributes to a given component. We noticed strong positive loading coefficients on PC2 for conventional chemotherapeutics, such as etoposide and paclitaxel (Fig 3D). To determine whether these observations were statistically robust, we quantified the statistical enrichment of each class of drugs on PC2, finding the strongest enrichments for topoisomerase inhibitors and microtubule poisons, two drug classes that are commonly used in the treatment of breast cancer (Fig 3E). We selected a subset of drugs to re-evaluate that had strong positive loading coefficients on PC2, suggesting that BL and ML cells would

have clearly distinct responses to these drugs. Consistent with these PCA generated insights, we found that BL cell sensitivity and ML cell sensitivity to etoposide and paclitaxel were clearly distinct in coculture, but not in monoculture, with BL cells generally being more sensitive to these conventional chemotherapies (Fig 3F and G). Taken together, these data demonstrate that coculture with fibroblasts was sufficient to enhance distinctions between BL and ML TNBC cells.

### Divergent interactions between TNBCs and fibroblasts based on the anatomical origin of fibroblast cells

Our coculture screen identified environmentally modulated drug responses that differed between BL and ML subtypes of TNBC. Notably, however, these distinctions were seen only for a small fraction of the drug responses tested in our screen, and primarily only in the context of drugs that are currently used in the treatment of TNBC. This distinction suggests that most of the response variation found in our screen is not associated with the BL/ML dichotomy and that opportunities may be found in our dataset that sensitize both BL and ML cells to conventional chemotherapeutics. Thus, we reasoned that deeper insights into the other sources of drug response variation in our data may reveal new strategies for potentiating drug responses in TNBC. PC1, which captured approximately 40% of the overall variation in drug sensitivity, was revealing a dominant pattern in our data that is unrelated to the BL/ML dichotomy (Fig 3A). To gain a deeper understanding of the sources of drug response variability in our data, we focused on identifying features of our data that were associated with PC1. We observed a noticeable clustering on PC1 for scores related to each fibroblast cell type (Fig 4A and Appendix Fig S9). To determine whether this pattern was revealing fibroblast cell-specific, or fibroblast tissue-specific variation, we computed the correlation between fibroblasts from the same tissue, versus between fibroblasts derived from different tissues. We found a significantly higher correlation between fibroblasts that were derived from the same anatomical tissue (Fig 4B and Appendix Fig S9). These data suggest that fibroblasts harvested from the same tissue modulate the drug responses of associated TNBC cells in similar ways.

Fibroblasts from different tissues could be modulating one of several aspects of TNBC drug response, including the response rate, magnitude of response (e.g., Emax or $EC_{50}$), or the directionality of the influence (e.g., increased or decreased sensitivity). Because our dataset included a surprising level of directional variability in fibroblast influence, we began by asking whether the directional variance in our screen could be attributed to distinct directionally specific influences of fibroblast from different tissues (*i.e.*, Is a fibroblast from a given tissue intrinsically more likely to induce drug sensitization/de-sensitization?). To test this, we calculated the ratio of drug responses for each TNBC–drug combination, in coculture versus monoculture. To facilitate visual inspection of the relative influences induced by each fibroblast type, we organized the data by dose and time, in order to highlight conserved fibroblast-dependent influences (Fig 4C). Each data tile was then subsequently grouped by stromal location and drug, and a map of this type was created for each TNBC cell line. From this analysis, clear differences between fibroblast lines were visible, with each fibroblast promoting either drug sensitization or desensitization (Fig 4D). These

patterns were similar between BL and ML subtypes, although ML TNBCs had more strongly polarized responses, in both positive and negative directions (Fig 4E). Some drug-specific responses, such as the desensitization of TNBCs to sunitinib, occurred regardless of which fibroblast was used in coculture. Interestingly, however, these drug-specific interactions were rare, and most fibroblast-induced changes in drug sensitivity were observed in a similar manner, across essentially all drugs, and in both BL and ML subtype TNBCs. Thus, rather than finding unpredictable or idiosyncratic interactions, specific to precise TNBC–fibroblast–drug combinations, our analysis reveals a dominant and relatively straightforward pattern: Fibroblasts modulate the drug response of TNBC cells in distinct and divergent manners, which are largely dependent on the anatomical origin from which the cells were harvested.

### Fibroblast-dependent, and drug-independent, variation in drug sensitivity occurs through modulation of the mitochondrial apoptotic priming state of TNBC cells

An unexpected phenotype from our screen was the degree to which a given fibroblast's influence over TNBC drug sensitivity was consistent regardless of which drug was applied (Fig 4D and E). Thus, we next aimed to determine the mechanism by which fibroblasts could interact with TNBC cells to produce these divergent—and largely drug identity independent—changes in TNBC drug response. The simplest mechanism that is consistent with this observation would be that these fibroblasts cause a direct modulation of TNBC cell growth or survival, independent of the drugs added (Fig 5A and Appendix Fig S10, example i). To test this, we used GFP-tagged TNBC cells to monitor TNBC-specific growth/survival phenotypes, in the absence of any drug. We found that most fibroblast cells either did not alter the growth rate of TNBC cells, or induced a modest growth rate increase in TNBC cells grown in coculture (Fig 5B). Furthermore, in the context of fibroblasts that consistently sensitized drug response of all TNBC cell lines (e.g., WS1, C12385, or HUF), coculture did not significantly decrease growth or survival, suggesting that a growth fitness or survival defect does not account for the broad-spectrum drug sensitization seen in coculture with these cells. In rare instances, coculture conditions did result in a significant TNBC cell growth rate decrease, such as seen with MDA-MB-231 cells grown with H6013, a fibroblast derived from lung tissue (Fig 5B). Notably, the M231-H6013 interaction induced broad-spectrum drug desensitization (*i.e.*, enhanced survival). Thus, even in the rare instances in which fibroblast cells mediated fitness defects, growth rate or survival modulation does not appear to account for the observed pattern of influences on TNBC drug response.

A second mechanism by which fibroblast cells could enhance drug efficacy could be by modulating drug bioavailability (Fig 5A, example ii). This could be achieved, for example, if fibroblast cells altered the levels of expression of drug efflux pumps in cancer cells, if fibroblasts sequestered drugs, or if fibroblasts metabolized the drugs, creating a more/less potent or bioavailable compound (Appendix Fig S10). This latter mechanism was recently reported to explain a microbiome–drug interaction that modulates toxicity of the chemotherapeutic 5-FU (García-González *et al*, 2017). To test the role of fibroblast modulation of drug metabolism or availability, we focused on drugs that activate

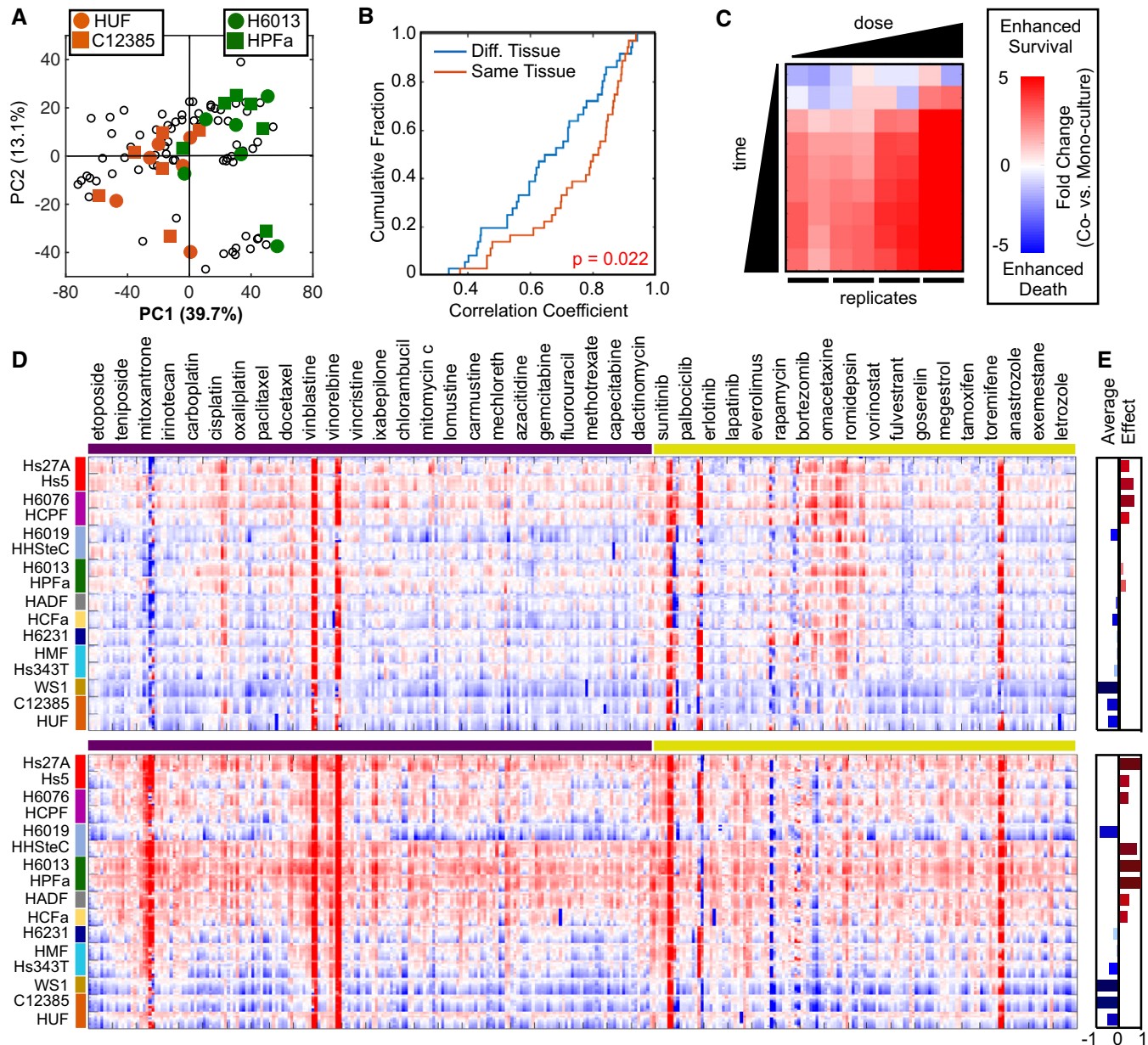

**Figure 4. Divergent interactions between TNBCs and fibroblasts based on fibroblast tissue of origin.**

A    PCA of coculture drug responses, as in Fig 3. Projection of scores shown to highlight separation of cocultures by fibroblast identity on PC1. Uterine-derived fibroblasts (orange) and lung-derived fibroblasts (green) highlighted. See also Appendix Fig S9.

B    Statistical analysis of fibroblast influence. Drug responses evaluated for fibroblasts from the same or different tissues using data from the coculture screen in Fig 2. *P*-value derived using KS test.

C, D    Ratio of coculture vs. monoculture drug response. 72 data points for each unique cancer–fibroblast–drug interaction arrayed by dose and time (nine time points and four doses in duplicate). (C) Example of total data for BT20-Hs27A-cisplatin. Data tiles are concatenated by fibroblast identity and drug in panel (D). (D, top) Average response ratio, as in panel (C), across all BL cell lines. (D, bottom) Average response ratio across all ML cell lines. 16 primary fibroblasts are grouped according to tissue of origin (highlighted by colored bar, left of heatmap). 24 left-most drugs (purple bar, top) are cytotoxic chemotherapies; 18 right-most drugs (yellow bar, top) are targeted therapies.

E    Average fibroblast influence across all drugs. Data are mean log$_2$ fold change (coculture vs. monoculture, as in D).

death through induction of DNA damage. For this set of compounds, drug potency should be proportional to level of $\gamma$−H2AX, which marks sites of DNA double-stranded breaks. We quantified $\gamma$−H2AX nuclear intensity following exposure to DNA-damaging agents, in the presence and absence of fibroblasts.

These measurements were made at four time points following exposure to 1 µM teniposide, cisplatin, etoposide, or camptothecin. We used GFP-labeled TNBC cells to identify TNBC cell nuclei, and images were quantified using a CellProfiler-based automated image analysis (Fig 5C; Lamprecht *et al*, 2007). We

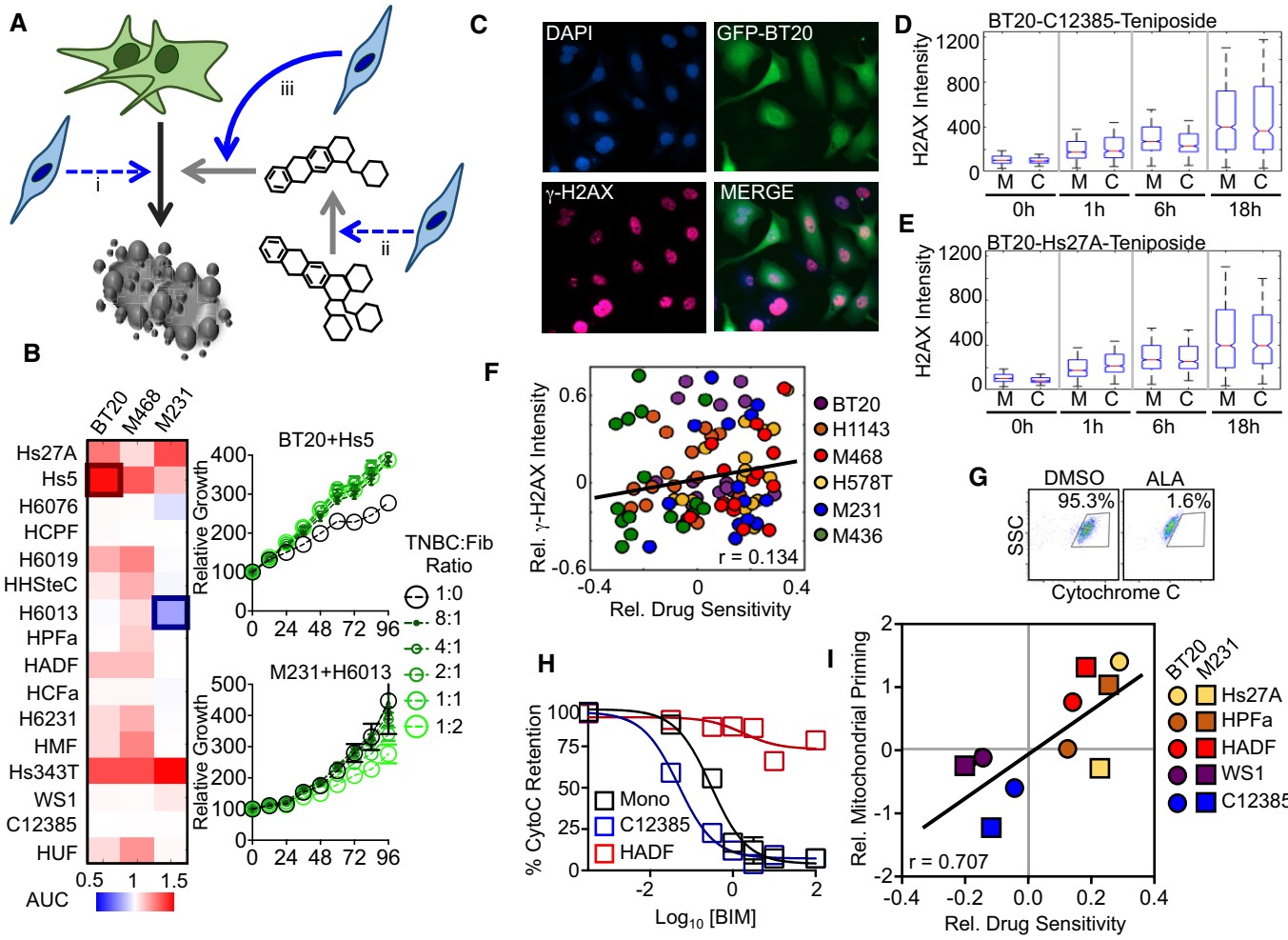

**Figure 5. Fibroblasts alter drug sensitivity through modulation of mitochondrial apoptotic priming.**

A   Schematic of possible mechanisms by which stromal cells alter drug sensitivity. See also Appendix Fig S10.
B   TNBC cells labeled with GFP were grown in monoculture or in coculture with listed fibroblasts. Growth rate quantified using a fluorescence plate reader. Heatmap data are area under curve (AUC) from biological triplicate measurements for cocultures plated at 1:1 cell ratio. Growth curves shown for the most enhanced and suppressed growth rate. Error bars are the standard deviation among four biological replicates.
C–F   γ-H2AX (p-H2AX, S139) monitored by immunofluorescence microscopy. (C) Representative image of GFP-BT20 cells cocultured with HADF. (D) Quantification of TNBC nuclear γ-H2AX from imaging experiment described in panel (C). In the boxplots, the red horizontal line represents the median; blue box show the 25th – 75th percentile range; whiskers show the full range of the data. Boxplot shown for BT-20 cells grown in monoculture (M) or in coculture (C) with strongly drug-sensitizing (C12385) fibroblasts. Cells counted using automated image analysis (CellProfiler). (E) As in panel (D), but for BT-20 cells grown in monoculture or in coculture with strong drug desensitizing (Hs27A) fibroblasts. (F) Scatterplot of coculture:monoculture viability ratio (from screen in Fig 2E) compared to coculture:monoculture ratio of γ-H2AX intensity. Both plotted in log₂ scale. For panels (D–F), average number of nuclei per counted per condition is 758 (range 93–1,632).
G–I   Evaluation of fibroblast influence on mitochondrial priming. (G) Cytochrome C retention quantified using the iBH3 profiling assay. Alamethicin (ALA) used as a positive control for mitochondrial rupture and cytochrome C release. (H) Mitochondrial priming quantified using exposure to varied concentrations of BIM in BT-20 cells. (I) Scatterplot comparing relative drug sensitivity (as in panel E) compared to degree of coculture induced change in mitochondrial priming. Priming status quantified as AUC from BIM dose response. Data are from biological quadruplicates. BT-20 (BL subtype); MDA-MB-231 (M231; ML subtype).

began by inspecting the most strongly sensitizing and desensitizing coculture environments to determine whether these extreme cases could be explained by differences in the apparent drug potency. TNBC nuclear γ−H2AX intensity was similar in BT-20 cells grown in monoculture or when cocultured with C12385, a uterine fibroblast that strongly sensitized TNBC drug response (Fig 5D). Similarly, differences in γ−H2AX between mono- and cocultures were also not observed when BT-20 cells were cocultured with Hs27A, a bone fibroblast that strongly desensitized

drug responses (Fig 5E). To more comprehensively determine whether modulation of drug potency could account for the observed pattern of drug sensitization/de-sensitization, we compared these changes in γ−H2AX intensity from quantitative image analysis, to the relative changes in drug sensitivity from our coculture screen. Overall, there was a low correlation between the degree to which γ−H2AX was modulated by fibroblasts and the phenotypic influence of these fibroblasts, which was not significant when compared to a randomized dataset

**Figure 6.  Modulation of mitochondrial priming by fibroblasts alters sensitivity to BH3 mimetic drugs.**

A  Representative images of GFP-BT20 cells grown in monoculture or in coculture with fibroblasts that increase or decrease the apoptotic threshold (C12385 and HADF, respectively). Each culture was treated with varying concentrations of the Topo II inhibitor teniposide or ABT737, BH3 mimetic that inhibits members of the BCL2 family of anti-apoptotic proteins. GFP-BT20 are green. In cocultures, fibroblast cells are red. In monocultures, BT-20 cells stained with a red cell dye were added to normalize plating densities.

B  Relative cell viability from experiment described in panel (A). Heatmap reports mean cell number, relative to untreated cells. Cells were quantified using CellProfiler from six images per drug condition. Average number of cells per image was 160. Each drug was tested at six doses, which were half-$\log_{10}$ dilutions from 10 µM.

C  Drug Combination Index (CI) relative to predicted % viability under Bliss Independence.

($r = 0.134$, $P > 0.05$; Fig 5F). Thus, it does not appear that fibroblast influences on drug sensitivity generally occur through modulation of drug availability or potency.

The insights gained from $\gamma$−H2AX intensity analysis are also consistent with our general observation that fibroblast cells influence drug sensitivity in similar ways across diverse and unrelated classes of drugs. In other words, it does not appear that the mechanisms by which fibroblast cells influence the drug responses in TNBC cells are specific to the drug compounds themselves or drug-specific responses of TNBC cells. Drug-induced cell death is the product of at least two independent influences: the drug-specific cell response (*i.e.*, the ability of a drug to change a cell from a healthy to a dead state) and the degree to which the cell is "primed" for death (*i.e.*, how "close" the healthy cell is to dying; Chonghaile *et al*, 2011; Montero *et al*, 2015). Thus, a third mechanism that we tested was whether fibroblasts alter the degree of mitochondrial apoptotic priming (Fig 5A, example iii). The state of apoptotic priming is thought to relate to the relative local concentration of pro- and anti-apoptotic proteins on the surface of mitochondria (Ryan *et al*, 2010). Thus, the priming state of a cell is not easily determined from gene or protein expression levels, but relative changes in priming can be empirically determined using the BH3 profiling technique (Ryan & Letai, 2013). This assay was recently used to demonstrate that normal or cancer-associated fibroblasts (CAFs) from mammary and brain tissue induce resistance to the HER2 inhibitor, lapatinib, in HER2 overexpressing breast cancers (Marusyk *et al*, 2016). Similarly, mammary- and brain-derived fibroblast also induced desensitization to lapatinib in our screen (Fig 4D). Thus, we used the BH3 profiling assay to assess the degree to which fibroblasts alter the mitochondrial priming state of TNBC cells. We selected five fibroblast cells that produced the strongest and most consistent modulation of drug sensitivity. The mitochondrial response to BIM peptide was quantified by monitoring cytochrome c retention within cancer cells by flow cytometry (Fig 5G). BH3 profiling revealed that fibroblast coculture significantly altered the mitochondrial priming state in both basal-like (BT-20) and mesenchymal-like (MDA-MB-231) TNBC cells (Fig 5H and I). Furthermore, the degree to which mitochondrial priming was increased or decreased was also highly correlated with relative drug sensitivity observed in our coculture screen (Fig 5I). Thus, in instances where fibroblasts strongly alter broad-spectrum drug sensitivities of TNBC cells, both the positive and negative changes in drug sensitivity are induced by modulation of mitochondrial priming.

We next aimed to determine whether the fibroblast-induced changes in TNBC mitochondrial priming translated to different levels of sensitivity to drugs designed to modulate the apoptotic threshold. For example, BH3 mimetic drugs function by inhibiting anti-apoptotic proteins, such as BCL2 and other related family members. Because these agents inhibit apoptotic inhibitors, but do not in and of themselves generate pro-apoptotic activating signals, their efficacy relies

on the native priming state of cancer cells (Adams & Cory, 2017). Thus, we reasoned that fibroblasts that enhance or suppress the priming state of cancer cells may differentially alter sensitivity to BH3 mimetic drugs. To test this, we exposed GFP-labeled BT-20 cells to varying concentrations of the topoisomerase II inhibitor teniposide and/or ABT-737, a broad-spectrum BCL2 family inhibitor. Images were collected following 48 hours of drug exposure (Fig 6A). To quantify cells, we used a CellProfiler-based automated image analysis. Contrary to what is commonly seen for many hematopoietic cancers, ABT-737 did not kill BT-20 cells when applied as a single agent (Fig 6B, *left*); however, combinations of ABT-737 and teniposide were generally synergistic when applied to BT-20 cells grown in monoculture (Fig 6C, *left*). The synergistic interaction between ABT-737 and teniposide was further enhanced when BT-20 cells were grown in coculture with C12385, a uterine fibroblast that enhanced apoptotic priming (Fig 6B and C, *middle*). In contrast, when BT-20 cells were grown in coculture with HADF, an adrenal fibroblast that potently decreased apoptotic priming, the drug synergy between these two agents was largely blocked, with most dose combinations resulting in additivity, or even modest drug antagonism (Fig 6B and C, *right*). These data confirm the role of fibroblasts in modulating apoptotic priming of associated tumor cells. Additionally, our data demonstrate that tumor–stroma interactions have the capacity to modulate, not only drug sensitivity, but also drug–drug interactions, further highlighting that these interactions are critical features that dictate how cancer cells respond to drugs.

## Discussion

In this study, we explored interactions between tumor cells and stromal cells to identify those that modulate sensitivity to commonly used chemotherapeutics. We found that fibroblasts alter drug sensitivity of tumor cells, and that the responses were highly variable, both in magnitude and in direction. Our statistical analysis clarified that the directional variability in fibroblast influence is predominantly associated with the anatomical tissue from which the fibroblast cells were harvested. This is an important observation, particularly considering that prior studies have typically only observed that stromal influences were unpredictable and/or very specific to the particular tumor–fibroblast–drug combination (Mcmillin *et al*, 2010). Surprisingly, we found that fibroblast-dependent changes in drug response were consistently observed, regardless of which drug was applied, which was driven by fibroblast-dependent modulation of mitochondrial apoptotic priming within cancer cells.

A major surprise from our work is the substantial directional variability in fibroblast influence and in particular the large proportion of tumor–fibroblast interactions that result in drug sensitization. Prior studies that have interrogated fibroblast–tumor

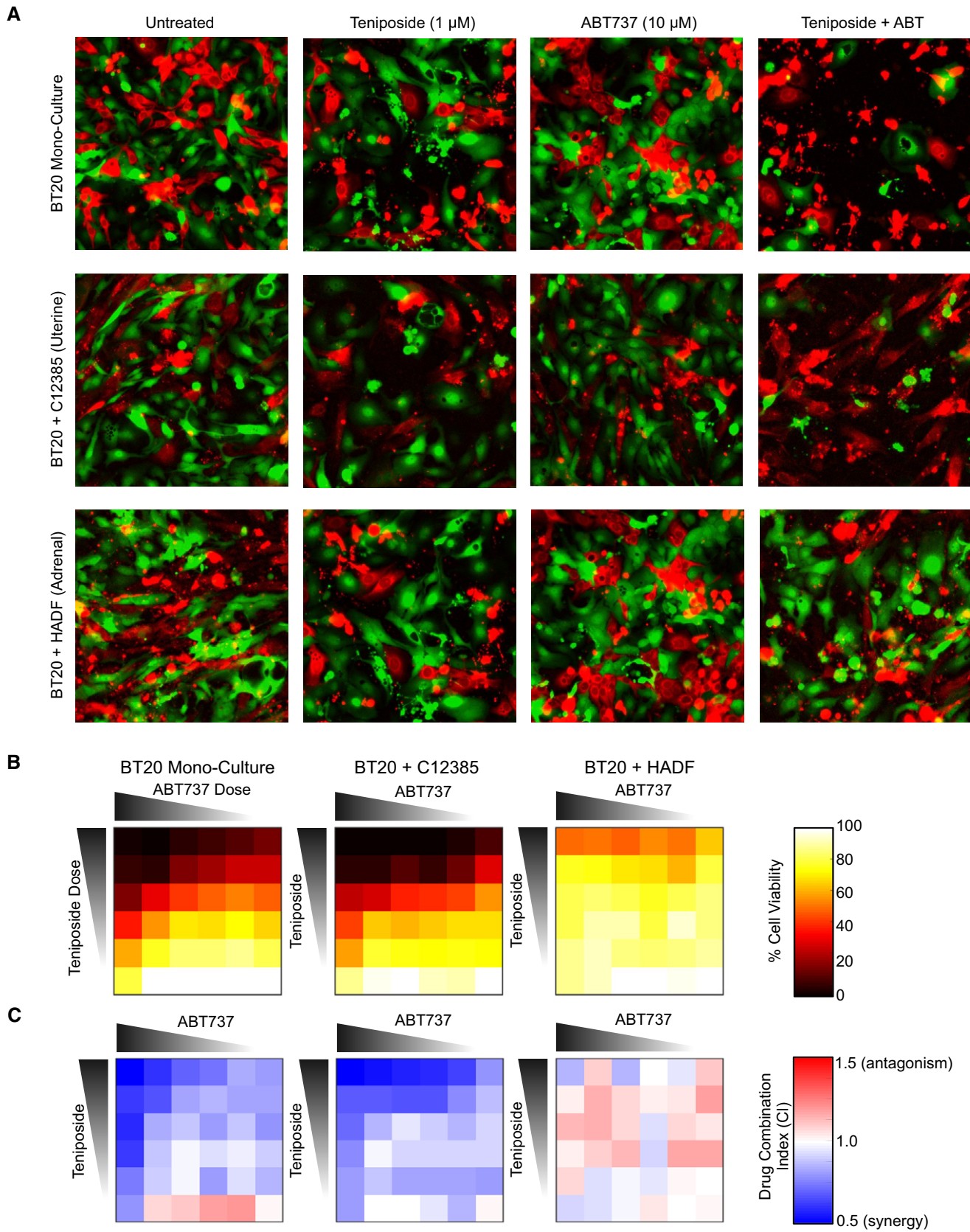

**Figure 6.**

cell–drug interactions have found that these interactions generally result in drug resistance, with only rare instances in which stromal cell interactions lead to drug sensitization (Mcmillin *et al*, 2010; Straussman *et al*, 2012; Marusyk *et al*, 2016). The differences observed in our screen likely resulted, in part, from the experimental scale used in our study. For example, some recent smaller scale studies have concluded that fibroblasts induce therapeutic resistance regardless of their tissue of origin (Marusyk *et al*, 2016). As noted above, our study successfully replicated the drug resistance phenotypes found in prior studies, and in our data, these drug resistance phenotypes are found in the context of many drug-sensitizing phenotypes that were not previously tested. An additional possibility is that our screening methodology, which was designed to exclusively monitor drug-induced cell death, contributed to the enhanced resolution of cell death sensitization. In fact, this feature is likely to have played a major role, considering the limited ability of other common approaches to quantify differences in the degree of cell death (Appendix Fig S2). A third possibility is that fibroblast-mediated drug sensitization is a more common phenotype in TNBC, as this cancer subtype was not deeply profiled in prior studies. A focused interrogation of other cancer subtypes may help to determine the extent to which fibroblast influences vary across different cancers.

Taken together, the findings from our study have potentially important implications that warrant consideration in the context of "personalized" or "precision" medicine, particularly for efforts to improve efficacy of commonly used therapies, such as cytotoxic chemotherapies or other pro-apoptotic agents. For instance, as we did not see differences in chemosensitivity between BL and ML cells grown in monoculture, it is possible that differences in chemosensitivity between these subtypes of TNBC may not be cell-intrinsic features, but instead may be the product of interactions between these cells and stromal cells. Thus, it is unclear whether detailed studies on the genomics of TNBC cells will be informative for identifying strategies to enhance chemosensitivity. Additionally, our experiments exploring the efficacy of BH3 mimetic compounds in combination with conventional chemotherapy reveal that the nature of drug–drug interactions (i.e., synergy or antagonism) depends strongly on the growth environment and not only on the cancer genotype. Furthermore, our ability to predict this drug–drug–environment interaction was facilitated by a mechanistic understanding of how fibroblasts modify drug responses in TNBC cells. Thus, future efforts to predict effective drug combinations *in vivo* will likewise require a greater understanding of how stromal cells from different tissues modulate the drug sensitivity of cancer cells.

Our findings highlight two potentially new opportunities to improve therapeutic responses in TNBC. First, our data show that the relative drug insensitivity of ML subtype TNBCs is restricted to only a few drug classes, which were generally strong apoptotic agents that are commonly used in treatment today. The drug specificity of the BL/ML dichotomy suggests that opportunities may exist for improving the responses of ML subtype cancers, perhaps in the context of other classes of drugs. Indeed, some drugs—which were equally efficacious in BL and ML cells grown in monoculture—were in fact more effective in the ML subtype when these cells were grown in coculture (see for example, bortezomib, Fig 4D). The ability to directly test these hypotheses *in vivo* is currently limited, as coculture xenograft models typically require the use of external

matrix (e.g., Matrigel) to support the implantation of fibroblasts, which obscures the tumor–fibroblast interaction (Appendix Fig S11). Second, the strong drug-independent influences of fibroblasts suggest that a more generalizable strategy may be to block interactions with drug desensitizing fibroblasts or to mimic the interactions of drug-sensitizing fibroblasts. Future studies should therefore aim to gain a comprehensive understanding of the mechanisms of interaction between fibroblasts and cancer cells, and in particular, the mechanisms by which fibroblasts alter the priming state of cancer cells.

## Materials and Methods

### Cell lines and reagents

Cell lines BT-20, HCC-1143, Hs578T, Hs578BST, MDA-MB-231, MDA-MB-436, MDA-MB-468, HCC-2157, HCC-1806, HCC-1395, Hs27A, HS-5, WI-38, IMR-90, Hs343T, WS-1 were obtained from American Type Culture Collection (ATCC, Manassas, VA), and cell line CAL-120 was obtained from Deutsche Sammlung von Mikroorganismen und Zellkulturen GmbH (DSMZ). All cell lines were grown in 10% FBS (Thermofisher Hyclone cat# SH30910.03 lot# AYG161519), 2 mM glutamine, and penicillin/streptomycin. BT-20, CAL-120, and WS-1 were cultured in Memα + Earle's salts. HCC-1143, HCC-2157, HCC-1806, and HCC-1395 were cultured in RPMI 1640 media. Hs578T, MDA-MB-231, MDA-MB-436, MDA-MB-468, Hs27A, HS-5, and Hs343T were cultured in Dulbecco's modified Eagle's medium (DMEM). Hs578T was further supplemented with 10 µg/ml insulin. Hs578BST was supplemented with 30 ng/ml EGF. Primary fibroblasts, H-6231, H-6201, H-6076, H-6019, and H-6013, were purchased from Cell Biologics (Chicago, IL); HCPF, HPF-a, HHSteC, HMF, HAdF, HUF, and HCF-a were purchased from ScienCell (Carlsbad, CA); and C-12385 was purchased from Promocell (Heidelberg, Germany). Primary fibroblast cells purchased from Cell Biologics, ScienCell, and Promocell were cultured in the media (ScienCell—Fibroblast Medium cat# 2301; Cell Biologics—Complete Fibroblast Medium/w Kit cat# M2267; Promocell—Fibroblast Growth Medium 2 cat# C-23020) for four doublings before being transitioned to DMEM. All cells were cultured at 37°C in a humidified incubator supplied with 5% $CO_2$ and maintained at a low passage number (< 20 passages for cancer). Prior to expansion and freezing, a small sample of each primary fibroblast was expanded to determine each cell's Hayflick limit to ensure that experiments could be performed prior to the onset of replicative senescence. A complete list of drugs used in this study is included in Table EV2.

### Coculture screen using JC1 dye

Fibroblast cell lines were grown to 80% confluence before being trypsinized and stained with 5 µM CellTrace Violet Proliferation dye (Thermofisher #C34557) in PBS at a concentration of $1 \times 10^6$ cell/ml for 15 min at 37°C. 1,500 stained cells were plated in 40 µl FluoroBrite media (Thermofisher # A1896701), supplemented with 10% FBS, 2 mM glutamine, and penicillin/streptomycin, in a Greiner clear 384-well plate (#781986) and allowed to adhere for 3 h. Cancer cell lines were then trypsinized and stained with 1.5 µg/ml

(final concentration) JC-1 (Thermofisher # T3168) in FluoroBrite at a concentration of $1 \times 10^6$ cell/ml for 20 min at 37°C. Cancer cells were then plated at 1,500 cells in 40 μl FluoroBrite per well in the 384-well plate. For monoculture conditions, unlabeled cancer cells were added to each well, in order to keep the cell density consistent with coculture conditions. Cells were allowed to adhere overnight. The following morning, 8 μl of a 10× drug stock was added to the wells using a VIAFLO 96 Electronic 96-channel pipetting robot. JC-1 fluorescence was then read at five spots across each well using a Tecan M1000 Pro Plate Reader at the excitation wavelength of 535 nM ± 17 nM and an emission wavelength of 590 nM ± 17 nM every 8 h for 72 h. Background fluorescence was determined by treating labeled cells with alamethicin, a membrane permeabilizing agent that punctures plasma membrane and mitochondrial membranes. Fluorescence measurements were normalized relative to pre-drug treatment values for each well.

## Cell viability and cell death assays

Cell viability assays were performed either using CellTiter-Glo (cat# G7570), for cells grown in monoculture, or flow cytometry, for coculture assays (other than the coculture screen, described above). For CellTiter-Glo, which measures viability as a function of ATP concentration, cells were plated in Greiner 96-well plates (cat# 655 090) at 5,000 cells per well in 100 μl of their respective growth media and allowed to adhere overnight. 10 μl of a 10× drug stock, diluted in PBS, was added to each well. Cells were subsequently allowed to grow at 37°C for 72 h. At 72 h post-drug addition, 33 μl of CellTiter-Glo reagent was added to each well. The CellTiter-Glo assay was performed according to manufacturer's directions, with the reagent diluted 1:3 (relative to media volume). Luminescence was read using a Tecan M1000 Pro Plate Reader. Cell death measurements to validate the JC-1 screen data were collected using the Live/Dead Violet reagent (Thermofisher cat# L34963) and analyzed by flow cytometry. Cancer cells and fibroblast cells were plated at a 1:1 ratio in DMEM and allowed to adhere overnight. Drugs were added from a 1,000× stock, and cells were exposed for the specified times. Cells were trypsinized at the specified times, suspended in PBS at a concentration of $1 \times 10^6$ cells/ml and stained with a 1:1,000 dilution of the Live/Dead Violet reagent for 30 min on ice. Cells were then fixed with 4% formaldehyde for 10 min at room temperature and run on an LSR II FACS machine with a laser excitation of 405 nm and emission of 450 nm.

## Drug sensitivity of cells grown in 3D culture conditions

Culturing of TNBC cells in 3D colonies was performed using the "3D on top" method developed by Bissell and colleagues (Lee *et al*, 2007). Briefly, a thick layer of cold Matrigel (corning cat#356235) was applied to the bottom of 96 well plates, which were subsequently heated to 37°C for 30 min to promote solidification. Cancer cells were plated at a concentration of 10,000 cells in 100 μl complete media + 2% Matrigel and were grown for 72 h to induce 3D colony formation. After 72 h of growth, the media were aspirated and media containing drug + 2% Matrigel were added to each well. At 72 h post-drug addition, cell viability was measured by adding 100 μl of CellTiter-Glo reagent to each well. The CellTiter-Glo assay was performed according to the manufacturer's

directions, and luminescence was read using a Tecan M1000 Pro Plate Reader.

## Growth rate measurements using GFP-labeled cells

To determine cell proliferation rate using a fluorescence plate reader, TNBC cells were stably transfected with GFP (pRetroQ-AcGFP1-N1). Transfected cells were selected with puromycin (BT-20 at 1.5 μg/ml, 468 at 0.5 μg/ml, and 231 at 2 μg/ml). Cells were selected until a parallel non-transformed plate exposed to puromycin was completely dead. The selected population was subsequently sorted by FACS to collect cells with similar levels of GFP fluorescence. For coculture experiments, fibroblast cell lines were plated at an 8:1, 4:1, 2:1, 1:1, and 1:2 ratio to cancer cells in a Greiner 96-well plate in 100 μl of FluoroBrite media and allowed to adhere for 3 h. Following adherence of fibroblast cells, TNBC cells constitutively expressing GFP were plated at a concentration of 10,000 cells per 100 μl of FluoroBrite media and allowed to adhere overnight. Cell measurements were measured every 24 h for 96 h using a Tecan M1000 Pro Plate reader.

## Fluorescence and immunofluorescence microscopy

For quantitative analysis of p-H2AX nuclear intensity, fibroblast cells were plated at a density of 1,500 cells per 25 μl in DMEM in a 384-well plate and allowed to adhere for 3 h. Cancer cells were stained with 5 μM CellTrace CFSE dye (Thermofisher cat# C34554) at a concentration of $1 \times 10^6$ cells/ml in PBS for 15 min at 37°C. Labeled cells were plated at 1,500 cells per 25 μl DMEM and allowed to adhere overnight. Drugs were added from a 10× stock solution in PBS, and cells were exposed for 1, 6, and 18 h before being fixed with 4% formaldehyde for 10 min at room temperature. Cells were washed twice in PBS, then permeabilized with 0.5% Triton X-100 for 10 min at room temperature. Cells were washed twice with PBS; blocked in 10% goat serum (Thermofisher cat# 16210064) for one hour; stained with the p-Histone H2A.X (Ser139) antibody (Cell Signaling Technologies #9718S) in 1% goat serum in PBS overnight at 4C; stained with Alexa-647 antibody (1:250 dilution, Thermofisher A21244) in 1% goat serum in PBS for 2 h at room temperature. Imaging was performed using an IXM-XL high-throughput automated microscope. Analysis was performed using a custom CellProfiler pipeline (available upon request). For imaging of GFP-labeled cells, cells were plated in 96-well plates at a density of 5,000 cells per well and allowed to adhere overnight. Following drug exposure (various times as indicated in figures), images were collected using an EVOS FL-AUTO automated fluorescence microscope. Analysis was performed using a custom CellProfiler pipeline (available upon request).

## SMA analysis by microscopy and FACS

Fibroblast cells were stained with 1uM CellTracker Deep Red Dye (Thermofisher cat#C34565) at a concentration of $1 \times 10^6$ cells/ml for 15 min at 37°C and subsequently plated in 96-well plates at a concentration of 10,000 cells per well and allowed to adhere for 3 h. Cancer cells were then stained with 5 μM CellTrace Violet Proliferation Dye at a concentration of $1 \times 10^6$ cells/ml for 15 min at 37°C and plated at a concentration of 10,000 cells per well and allowed to

adhere for 24 h. After 24 h, cells were fixed with 4% formaldehyde for 10 min at room temperature and washed twice with PBS. Cells were then permeabilized with 0.5% Triton X for 10 min at room temperature, washed twice with PBS, and then blocked in 10% goat serum for one hour. Cells were stained with the αSMA antibody (1:75 dilution, Cell Signaling Technology cat#19245S) in 1% goat serum for 8 hours at room temperature and then stained with the Alexa-488 antibody (1:100 dilution Thermofisher cat#A11008) in 1% goat serum for 1 h at room temperature. Imaging was performed using an EVOS FL-AUTO automated fluorescence microscope. A similar process was used for analysis of SMA expression by FACS, except following formaldehyde fixation, and cells were exposed to 100% methanol for 2 h at −20°C and washed twice with PBS. Cells were stained with the αSMA antibody (1:75 dilution, Cell Signaling Technology cat#19245S) in 50% Odyssey Blocking Buffer (LI-COR cat# 927-40000) 50% PBS-T for 8 hours at room temperature and then stained with the Alexa-488 antibody (1:100 dilution Thermofisher cat#A11008) for 1 h at room temperature in 50% Odyssey Blocking Buffer 50% PBS-T. FACS was run on an LSR II machine with a laser excitation of 488nm and emission of 530 nm.

### Mitochondrial priming assay

Mitochondrial priming assays were performed according to the iBH3 protocol from Ryan *et al* (2016). For monoculture conditions, $1 \times 10^6$ cancer cells were plated in a 10-cm dish and allowed to adhere overnight. For coculture conditions, fibroblasts were stained with Cell Trace Violet plated at a 1:1 ratio with cancer cells. Twenty-four hours post-plating cells were trypsinized and arrayed in a 384-well plate. A dose series of BIM peptide (GenScript) was added (100, 33, 10, 3.3, 1, 0.33 μM) along with either DMSO (vehicle control) or alamethicin (Enzo, BML-A150-0005), a mitochondrial depolarizing agent, which was used at a final concentration of 25 μM as a positive control. Plasma membrane permeabilization was achieved by the addition of digitonin at a final concentration of 20 μg/ml. Cells were incubated with BIM peptide at room temperature for 1 hour before being fixed and stained for cytochrome c retention (Fisher cat# BDB560263). Samples were analyzed on an LSRII flow cytometer.

### Data analysis and statistics

All statistical analyses were performed using GraphPad Prism and/or MATLAB, generally using pre-built functions (Fisher's exact test, *t*-test, etc.). PCA was performed using SIMCA, and data were *z*-scored (mean centered and unit variance scaled). Hierarchical clustering was performed using Spotfire using the default settings (UPGMA clustering method; Euclidean distance measure; average value ordering weight; *z*-score calculation normalization method; empty value replacement: NA). Analysis of flow cytometry data was performed using FlowJo. Combination Index (CI) was calculated by dividing observed drug sensitivity by the expected drug response given Bliss Independence (for two drugs, A and B, expected = A + B − [A*B]).

### Data availability

Coculture screening data are provided in Table EV3. The CellProfiler scripts used for the analyses shown in Appendix Fig S2, Figs 5, and 6 are provided as Code EV1, Code EV2, and Code EV3, respectively.

**Expanded View** for this article is available online.

### Acknowledgements

We thank members of PSB and the DFCI CCSB for valuable discussions during the design and execution of this study. Additionally, we thank the UMass Med School Flow Cytometry Core and High-Throughput Imaging Core for training and advice during the execution of this study. Research reported in this publication was supported by the National Institute of General Medical Sciences of the National Institutes of Health under Award Number R01GM127559 (MJL) and by grants to MJL from the Richard and Susan Smith Family Foundation, Newton, MA; the Breast Cancer Alliance; the American Cancer Society (RSG-17-011-01). BDL, PCG, and RR were supported by a NIH training grant (Translational Cancer Biology Training Grant, T32-CA130807). Additional support was provided by the UMass President's S&T Fund to MJL and SRP.

### Author contributions

This project was conceived by BDL and MJL. Experiments were designed by BDL, HRS, ADS, SRP, and MJL. Coculture screen was designed and executed by BDL. All other experiments were executed by BDL, TL, RR, PC-G, HRS, and GR. Analysis was conducted by BDL, TL, RR, HRS, MEH, and MJL. Manuscript was written and edited by BDL, ADS, SRP, and MJL.

### Conflict of interest

The authors declare that they have no conflict of interest.

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
