## [Review Process File · Molecular Systems Biology]

Tumor-Stroma Interactions Differentially Alter Drug Sensitivity Based on the Origin of Stromal Cells

Benjamin D. Landry, Thomas Leete, Ryan Richards, Peter Cruz-Gordillo, Hannah R. Schwartz, Megan E. Honeywell, Gary Ren, Alyssa D. Schwartz, Shelly R. Peyton, Michael J. Lee.

Review timeline:

Submission date:	16 th March 2018
Editorial Decision:	12 th April 2018
Revision received:	21 st June 2018
Editorial Decision:	10 th July 2018
Revision received:	15 th July 2018
Accepted:	19 th July 2018

Editor: Maria Polychronidou.

Transaction Report:

1st Editorial Decision

12th April 2018

Thank you again for submitting your work to Molecular Systems Biology. We have now heard back from the two reviewers who agreed to evaluate your manuscript. As you will see below, the reviewers raise substantial concerns on your work, which unfortunately preclude its publication in Molecular Systems Biology.

The reviewers appreciate the extensive amount of data generated in the study. However, reviewer #1 points out that as it stands, several of the conclusions are not convincingly supported. As such, s/he indicated that s/he does not support publication of the study in Molecular Systems Biology. While reviewer #2 is more positive, s/he does mention that the absence of some level of mechanistic insight or evidence that the main findings are relevant *in vivo* represents a limitation. As such, at this point we see no choice but to return the manuscript with the message that we cannot offer to publish it.

However, since the reviewers do acknowledge that the topic of the study is relevant and the main findings seem potentially interesting, we would not be opposed to considering an extended manuscript based on this work, provided that the issues raised by the reviewers can be convincingly addressed. In particular, further analyses should be included i) testing further cell lines per subtype to provide better support for the related conclusions, ii) demonstrating the relevance of the findings for cancer-associated fibroblasts, iii) providing support for the *in vivo* implications of the main findings. We recognize that thoroughly addressing the referees' concerns would involve substantial further experiments and analyses with unclear outcome. Of course, I understand if in light of the substantial revisions required, you prefer to submit your study elsewhere.

A resubmitted work would have a new number and receipt date and as you probably understand, we can give no guarantee about its eventual acceptability. If you do decide to follow this course then we would ask you to enclose with your re-submission a point-by-point response to the points raised in

the present review.

REFERE REPORTS.

Reviewer #1:

The authors of this manuscript conducted a large-scale co-culture screen to study the influence of stromal cells on the sensitivity of triple negative breast cancer (TNBC) cells to a wide range of commonly used chemotherapeutic agents. They show that this co-culture state is more indicative of phenotypes seen in patients than 2D or 3D monocultures. They also show that fibroblasts influence the tumor cells response to drugs independent of the tumor subtype or the target of the drug. This is a descriptive study with limited novelty, since numerous papers have published that stromal cells influence therapeutic responses in breast and other cancer types and several of these past papers included similar screens. The numbers of stromal cells used in the screen, the dataset, and the finding that fibroblasts behave similarly despite the drug and cancer cell type, is a meaningful contribution to the field. However, several of the conclusions are not supported by the data.

Specific points:

1. Figure 1/Supp Figure 1: The use of 10 cell lines is not sufficient to prove that subtype does not impact responses to treatment. The authors have to test at least 10 cell lines/subtype. BT20 and HCC2157 have some luminal features, so they are not clear basal cells. The authors should also indicate on each figure what the subtype is (like use different colors for basal and mesenchymal).
2. Prior publications have described that basal subtype is associated with better response to chemotherapy in patients.
3. The authors used fibroblasts from different normal organs, but not cancer-associated fibroblasts. This is a major caveat, since cancer-associated fibroblasts behave very differently from normal ones.
4. In Figure 1C what are the numbers for BL and ML? Can this also be shown as a heatmap with hierarchical clustering?
5. In the conclusion of Figure 1, the authors state that the ML and BL subtypes are similarly sensitive when cultured in mono-culture in 2D or 3D. This is an overstatement as the trend of ML being more resistant is still present.
6. Please comment on the rationale as to why the plate reader would fail to capture this significant loss of GFP expression seen in the camptothecin sample. Is this due to the intensity of the GFP expression or the fluorophore? If other fluors were used to mark the cells would there still be this discrepancy?
7. Figure 2E: More information could be gained from this graph if it was a density plot like others in the supplement. Also, how do the individual time points change over time? Do they generally increase in severity over time?
8. Supplemental figures 4-6: overall dataset in blue is busy and distracting. Highlight significantly different interactions.
9. Supplemental figure 6: Include origin of fibroblasts in parentheses next to the fibroblast name to make it easier to visualize trends.
10. In Figure 3 the authors show there are differences between classes of drugs in BL and ML treatment in co-culture. What is the mechanistic explanation for this?
11. In Figure 4a: what about the other classifications of fibroblasts?
12. Figure 5b is surprising based on literature, are any of the results generated by the screen

reproducible when cultured in 3D?

13. Figure 5D is unclear. Plot individually?

14. Overall the figure legends need to be more specific and clear as to how the data is generated. For example, in Figure 2F-I and Figure 4b

Reviewer #2:

In this manuscript the authors explore the impact of co-culturing triple negative breast cancer cell lines with primary fibroblasts derived from diverse anatomical locations. To set the stage the authors demonstrate the relative shortcomings of monoculture formats, both 2D and 3D, to distinguish relatively chemo-sensitive basal-like and relatively chemoresistant mesenchymal-like cell states in drug treatment assays. The association between these cell states and chemo-sensitivity/resistance is well-established clinically yet difficult to reproduce in culture systems that could be used to identify efficacious new treatments. The authors rectify this unmet challenge by creating a co-culture system of cancer cells and primary fibroblast. Additionally, they develop a new cell-based screening approach that focuses on measuring cell death opposed to cell proliferation and demonstrate its superiority over traditional methods. With this system, the authors co-culture 6 TNBC cell lines with 16 different fibroblast cultures in the presence of a four point dose escalation of 42 anti-cancer compounds. In total they made over 300,000 measurements. Besides the impressive scale, the authors made several high impact findings:

- * The study was made possible by the author's development of a sensitive high-throughput assay to measure cell death using a fluorescent plate reader; other methods were not sensitive enough to detect cytotoxic effects that were obvious upon microscopic analysis.
- * 5,039 differential drug responses were detected when comparing mono- and co-culture of TNBC cells and diverse fibroblasts.
- * The differential sensitivity of basal-like and mesenchymal-like cells could be captured by co-culturing with fibroblasts. This is a significant development as it has the potential to identify the molecular differences that underlie differential sensitivity and thus be exploited for better therapeutic approaches. *See Note below.
- * Fibroblasts from different tissues affected the sensitivity to chemo in distinct ways. This finding sets the stage to contrast the molecular underpinnings of these differential effects. *See Note below.
- * Differences in drug sensitivities when co-cultured were surprisingly consistent across most drugs screened. This is surprising because it intimates that the effects of the fibroblast are quite general to TNBC cell physiology and insights made that inform treatment strategies could be applied to a large fraction of TNBC patients.
- * Finally, the authors describe mitochondrial cell death priming mechanism to explain the differences observed in drug sensitivity during co-culture. Though the depth of this mechanistic insight is limited the results do portend a general approach that could be used to sensitize TNBC to current treatments. Namely, BH3 mimetics are clinically relevant drugs and the connection between fibroblast effects and mitochondrial priming suggest that alternative methods to change mitochondrial priming could be attained by modulating the fibroblast. Ostensibly this indirect approach could alleviate some untoward side effects of BH3 mimetics.

Overall impression: This manuscript is very well-written, easy to interpret and takes on an important problem in cancer treatment using a systems based approach. Several novel-insights were made and innovative new approaches were developed and implemented with a high degree of experimental rigor. *The study does lack significant mechanistic insights. The idea that mitochondrial priming is altered is interesting but how that occurs in one co-culture versus another is not explored. Clearly these authors are well-positioned to identify those factors that drive these differences. The question is whether that should be explored in more detail here. Clearly, identifying the precise effector molecules would raise the impact of this manuscript, but it would also likely make it more appealing to journal with a larger readership. However, perhaps some further demonstration of how a key finding within the authors' co-culture dataset could also show increased efficacy in an in vivo model would be appropriate and still be within the scope of this publication.

We thank the reviewers for their thoughtful comments on our work. Detailed responses are included below (blue font) in the context of the reviewer's comments (black font).

Reviewer #1:

The authors of this manuscript conducted a large-scale co-culture screen to study the influence of stromal cells on the sensitivity of triple negative breast cancer (TNBC) cells to a wide range of commonly used chemotherapeutic agents. They show that this co-culture state is more indicative of phenotypes seen in patients than 2D or 3D monocultures. They also show that fibroblasts influence the tumor cells response to drugs independent of the tumor subtype or the target of the drug. This is a descriptive study with limited novelty, since numerous papers have published that stromal cells influence therapeutic responses in breast and other cancer types and several of these past papers included similar screens. The numbers of stromal cells used in the screen, the dataset, and the finding that fibroblasts behave similarly despite the drug and cancer cell type, is a meaningful contribution to the field. However, several of the conclusions are not supported by the data.

We thank the reviewer for highlighting that the large dataset and central findings are meaningful contributions to the field. Regarding the lack of novelty, we feel that our study has several areas of novelty, many of which were highlighted by Reviewer #2. Most critically, it may be worth emphasizing that our main finding is *not* simply that fibroblasts alter drug responses. As stated by the reviewer, this fact has indeed been highlighted by prior studies, sometimes using approaches similar to ours. Our main finding is that fibroblasts both sensitize and de-sensitize TNBC drug responses (the former has not been well reported) and that these polarized responses are associated with the tissues fibroblast tissue of origin. We think that this distinction is important as it places stromal cells in a central role in shaping how tumor cells respond to drug therapy. These findings should influence how the community thinks about “personalized” or “precision” medicine, as the current conversation is focused entirely on tumor genomics.

Specific points:

1. Figure 1/Supp Figure 1: The use of 10 cell lines is not sufficient to prove that subtype does not impact responses to treatment. The authors have to test at least 10 cell lines/subtype.

In principle, we agree with the reviewer that a small sample of cell lines would not be sufficient to prove that tumor subtype does not impact responses to treatment. However, we think there appears to be a miscommunication with regards to our interpretation of the data in Figure 1. It is very much our assumption that tumor subtype does impact the response to treatment, or at least that this *should* be true since it is true in patients. The data in Figure 1 simply highlight that the expected subtype dependent differences in drug response are generally not observed *in vitro*. This is true in our data, and also true in publically available data collected by the NIH/NCI LINCS Consortium. Based on this observation, our study explores the hypothesis that the subtype dependent responses to treatment are the product of tumor-stroma interactions. Our screening data and statistical modeling using PCA support this hypothesis, as the expected patterns of subtype dependent treatment response were observable when TNBCs were co-cultured with fibroblasts. We apologize for this confusion and have rewritten several portions of the text to better highlight that our *in vitro* data are not meant to challenge the clinical observation that BL and ML subtype TNBCs differ in their drug sensitivity (see for example final sentences of the first paragraph of page 6 and 7).

Additionally, as the reviewer suggested, our conclusions would be more robust if at least 10 BL and 10 ML cells were tested. We apologize again that this was not entirely clear, but the LINCS consortium data that was used in Figures 1C was generated from 24 TNBC cell lines, 11 of which are basal-like and 13 of which are mesenchymal-like. This point has been added to both the text associated with Figure 1C and the figure legend.

BT20 and HCC2157 have some luminal features, so they are not clear basal cells.

We were using the consensus designations from the NIH/NCI funded LINCS Consortium (Library of Integrated Network-based Cellular Signatures, <http://www.lincsproject.org>; <http://lincs.hms.harvard.edu>), which is a multi-institution effort to collect genome-based information to that describes how cancer cells respond to stresses, such as common drugs. Both BT20 and HCC2157 are described as Basal A Subtype. This is similar to what was found in large studies

conducted by Jennifer Pietenpol and Joe Gray (Lehmann et al., JCI 2011; Neve et al. Cancer Cell 2006).

The authors should also indicate on each figure what the subtype is (like use different colors for basal and mesenchymal).

We thank the review for this suggestion. It appears that this information was available on Figure 1, but not consistently available in subsequent figures. We have added subtype information for each cell line in the figure legends of subsequent figures.

2. Prior publications have described that basal subtype is associated with better response to chemotherapy in patients.

As highlighted above under Specific Point #1, we are in agreement that this should be the expected response and have modified the text to make this point more clear.

3. The authors used fibroblasts from different normal organs, but not cancer-associated fibroblasts. This is a major caveat, since cancer-associated fibroblasts behave very differently from normal ones.

This is an important point, and we regret not addressing this issue more clearly in the initial submission. Our initial point of view was that primary fibroblasts and CAFs (while indeed different in many respects) were similar in their ability to modulate drug responses in cancer cells. This was addressed indirectly in a study by Straussman and Golub (Nature 2012), which used primary fibroblasts to identify mechanisms of resistance to BRAF inhibition *in vivo*. Additionally, Kornelia Polyak's group has recently tested this issue directly and found that fibroblasts derived from breast/brain tissue and or breast/brain cancers were all similar in their ability to desensitize breast cancers to commonly used drugs (Marusyk et al. Cancer Research 2016). In some important aspects, our data (and more importantly, our interpretation of our data) are not fully consistent with interpretations from the Marusyk et al. study, so we agree that it is indeed critical that we also address the similarity of our primary fibroblasts to *bona fide* CAFs (**see note below).

In this revised manuscript, we included two new pieces of data that address the similarity of our primary fibroblasts to CAFs, and these data are shown in Supplemental Figure 7. First, we tested our fibroblasts for markers of "activation". We chose α -smooth muscle actin (SMA), which is commonly used as a marker of CAFs (Shiga et al. 2015). Our primary fibroblasts were generally activated even when grown in mono-culture (median percent SMA+ fraction was 63%) and the SMA positivity was consistently increased by co-culture with basal or mesenchymal cells. The high levels of fibroblast activation in culture have been observed previously, and are reported to be a reaction to the stiffness of standard cell culture environments (Huang et al. 2012). These data and related citations are highlighted in the manuscript in page 10.

Secondly, we also directly addressed the similarity of primary and CAF cells, at least with respect to their ability to modulate drug sensitivity of TNBC cells. We focused on testing breast cancer associated fibroblasts. This effort was limited by the commercial availability of CAFs, particularly those that can be expanded without undergoing senescence (we purchased two breast cancer derived CAF from a commercial vendor, but these ultimately could not be cultured to sufficient numbers without reaching their Hayflick limit). We were able to test two breast CAFs, Hs343T and Hs578BST, which could be compared to HMF, a normal primary breast fibroblast. Note that one of these CAFs, Hs343.T, was included in our original screen. This cell had been characterized as a primary breast fibroblast based on its characterization in a prior study (Straussman et al. Nature 2012). Based on the documentation from ATCC, this is indeed a CAF derived from a patient with breast adenocarcinoma. In our retest of HMF and Hs343T, the data were very similar to our original screening data (comparison of boxplot to dots in Sup Figure 7D). Importantly, Hs578BST – a CAF tested here for the first time that was derived from the same tumor as Hs578T – modulated sensitivity to common drugs in a manner that was similar to Hs343T and HMF. Overall, we observed a significant correlation between CAF induced changes in drug sensitivity and those induced by the primary fibroblast HMF. Clearly, our ability to deeply interrogate the relationship between primary fibroblasts and CAFs has been limited by the availability of appropriate material. We hope that the reviewer will consider this a "good faith" effort and also consider that this issue was addressed by the Polyak group as well. We were able to confirm the finding from the Marusyk et al. study, which reported that primary and cancer associated fibroblasts appear to play similar roles in modulating drug sensitivity of associated cancer cells.

***To be clear about the discrepancy between our interpretations and those in the Marusyk et al. study, although our data recapitulate theirs (breast cancer cells were consistently desensitized to lapatinib when co-cultured with mammary or brain derived fibroblasts), we come to a different conclusion due to the scale/scope of our study. In the Marusyk et al. paper the authors conclude that fibroblasts desensitize cancer cells to drugs, regardless of their anatomical origin and regardless of their primary or CAF status. Due to the larger scale of our study (many more drugs, and more importantly, many more fibroblasts) our data reveal that fibroblasts from different tissues do indeed alter drug sensitivity of cancer cells differently (Figure 4).

4. In Figure 1C what are the numbers for BL and ML? Can this also be shown as a heatmap with hierarchical clustering?

We apologize that this wasn't made more apparent. The LINCS data has 11 BL and 13 ML cells. Also, as the reviewer suggested, we have also added a heatmap with hierarchical clustering for these data in Supplemental Figure 1D.

5. In the conclusion of Figure 1, the authors state that the ML and BL subtypes are similarly sensitive when cultured in mono-culture in 2D or 3D. This is an overstatement as the trend of ML being more resistant is still present.

We can see the reviewer's point, for example with the drug camptothecin or topotecan. These trends, however, were not statistically significant. In fact, for all drugs tested, the differences in sensitivity of BL and ML cells were not statistically significant, both in 2D and also in 3D. We have added a sentence to highlight this in the text on page 6.

6. Please comment on the rationale as to why the plate reader would fail to capture this significant loss of GFP expression seen in the camptothecin sample. Is this due to the intensity of the GFP expression or the fluorophore? If other fluors were used to mark the cells would there still be this discrepancy?

The issue appears to be the stability of GFP in media after cells have died. We added data from our original titrations, which address this point (Supplemental Figure 2D). In this experiment, we plated cells at 2-fold serial dilutions to determine the dynamic range of the GFP assay. Fluorescence was clearly above background when cells were plated > 1200 cells/well (our assay typically used 5-10K cells per well in a 96 well plate). Thus, we were somewhat surprised to see that the signal did not decrease when cells died. Shown in the figure, for example, is 0.2% Triton-X, which kills all cells within 20 minutes (no cells are visible by microscopy). The addition of Trypan Blue, which quenches GFP fluorescence but does not enter intact cells, did reduce Triton-X permeabilized cells to background fluorescence but did not change fluorescence of healthy cells. This suggests that GFP molecules that are released into media following cell death still retain fluorescent properties long after cell rupture. We had not previously tested other fluors but did try mKATE to address this concern. The mKATE fluorescence was much lower than that of GFP but still above background at high cell densities. Notably, even 8 hours after cell lysis with Triton the mKATE signal is unchanged.

7. Figure 2E: More information could be gained from this graph if it was a density plot like others in the supplement. Also, how do the individual time points change over time? Do they generally increase in severity over time?

For this figure, we aimed for simplicity, as not to make it too busy. As one might expect, overall there is a trend towards more death with increasing time and increasing dose, but there isn't a single dose or time that adequately captures the trends for all cells/drugs. Each instance (e.g. drug/cell/fibroblast combo) has an unpredictable saturation point. Information would be lost if we picked a single dose or time to "simplify" the data. To add the requested information, we created a new scatter plot that highlights cell line, dose, and time. This plot has been added to a revised Supplemental Figure 3.

8. Supplemental figures 4-6: overall dataset in blue is busy and distracting. Highlight significantly different interactions.

We tried several iterations of new figures in place of Supplemental figures 4-6. Removing the blue (which represent the overall data) often impaired our ability to see subtle but global shifts in the data. For example, the HCC1143 cell line is largely shifted towards the lower right, but this was less obvious without the aid of the blue dots. We also tried highlighting significantly changed drug

responses as in Supplemental Figure 3E, but this felt too distracting. In place of adding this information to the figures, we have supplied a new Supplemental Table 4 which highlights the number of significantly changed drug responses for each cell line, drug, fibroblast, etc.

9. Supplemental figure 6: Include origin of fibroblasts in parentheses next to the fibroblast name to make it easier to visualize trends.

The requested information has been added.

10. In Figure 3 the authors show there are differences between classes of drugs in BL and ML treatment in co-culture. What is the mechanistic explanation for this?

The PCA plot (Figure 3A) shows that the BL/ML dichotomy is only associated with PC2 implying that a subset of drug responses were distinct between BL and ML cells. In conjunction with this, the loadings plot (Figure 3D) shows that differences between BL and ML cells were observed mainly for 4 classes of drugs (Figure 3E). The mechanistic explanation for this is likely that these classes of drugs elicit high levels of apoptotic cell death, whereas the other classes were either inefficacious, inducing growth arrest, and/or non-apoptotic death. This is consistent with our later observation that fibroblasts mediate their effects by modulating the apoptotic threshold. A sentence has been added to the discussion on page 20 to highlight this point.

11. In Figure 4a: what about the other classifications of fibroblasts?

The other fibroblasts are shown in this manner (PCA scores plot) in Supplemental Figure 8, and the comprehensive statistical analysis of these tissue specific interactions is shown in Figure 4B. We apologize about this confusion, and the information has been added to the text on page 13.

12. Figure 5b is surprising based on literature, are any of the results generated by the screen reproducible when cultured in 3D?

Figure 5B highlights that most fibroblasts have a modest but positive affect on the growth rate of cancer cells in co-culture. These are relatively small trends and are highlighted here to negate the possibility that the apparent drug desensitization phenotypes were due solely to slowed growth or enhanced death due to fibroblast cells (rather than the drugs, per se). This information has been better highlighted in the text on page 15. Regarding whether these subtle phenotypes would be seen in 3D, this would be difficult to test directly, particularly as the growth phenotypes are relatively small. Our 3D culture experiments in Figure 1 were performed by embedding TNBC cells in matrigel. As shown in a new Supplemental Figure 11, matrigel is itself an environment that causes changes in the behaviors of TNBC cells and the matrigel-induced phenotypes tend to be dominant.

13. Figure 5D is unclear. Plot individually?

Figure 5D (now Figure 5F) shows the overall correlation between the fibroblast induced change in p-H2AX (a marker of DNA double stranded breaks) and fibroblast induced change in drug sensitivity. These data are overall averages from our co-culture screen and from the imaging experiment described in Figure 5C. The main point is only that there is no correlation between these features (i.e. that changes to the potency of DNA damaging drugs does not account for observed changes in the drug efficacy). The boxplots that are now shown in Figure 5D and E do a better job of explaining our main message. These show that in two extreme cases (strong fibroblast induced drug sensitization and strong desensitization) we observed no change in H2AX levels. We don't think it would be helpful to plot the cell lines (or the two axes of the scatter plot) individually, but we have re-written the text and re-organized the figure to de-prioritize the scatter plot and prioritize the boxplots that demonstrate that levels of DNA damage are not strongly altered by the presence of fibroblasts.

14. Overall the figure legends need to be more specific and clear as to how the data is generated. For example, in Figure 2F-I and Figure 4b

We have re-written the figure legends to add more specificity regarding data generation methods.

Reviewer #2:

In this manuscript the authors explore the impact of co-culturing triple negative breast cancer cell lines with primary fibroblasts derived from diverse anatomical locations. To set the stage the authors demonstrate the relative shortcomings of monoculture formats, both 2D and 3D, to distinguish relatively chemo-sensitive basal-like and relatively chemoresistant mesenchymal-like cell states in

drug treatment assays. The association between these cell states and chemo-sensitivity/resistance is well-established clinically yet difficult to reproduce in culture systems that could be used to identify efficacious new treatments. The authors rectify this unmet challenge by creating a co-culture system of cancer cells and primary fibroblast. Additionally, they develop a new cell-based screening approach that focuses on measuring cell death opposed to cell proliferation and demonstrate its superiority over traditional methods. With this system, the authors co-culture 6 TNBC cell lines with 16 different fibroblast cultures in the presence of a four point dose escalation of 42 anti-cancer compounds. In total they made over 300,000 measurements. Besides the impressive scale, the authors made several high impact findings:

* The study was made possible by the author's development of a sensitive high-throughput assay to measure cell death using a fluorescent plate reader; other methods were not sensitive enough to detect cytotoxic effects that were obvious upon microscopic analysis.

* 5,039 differential drug responses were detected when comparing mono- and co-culture of TNBC cells and diverse fibroblasts.

* The differential sensitivity of basal-like and mesenchymal-like cells could be captured by co-culturing with fibroblasts. This is a significant development as it has the potential to identify the molecular differences that underlie differential sensitivity and thus be exploited for better therapeutic approaches. *See Note below.

* Fibroblasts from different tissues affected the sensitivity to chemo in distinct ways. This finding sets the stage to contrast the molecular underpinnings of these differential effects. *See Note below.

* Differences in drug sensitivities when co-cultured were surprisingly consistent across most drugs screened. This is surprising because it intimates that the effects of the fibroblast are quite general to TNBC cell physiology and insights made that inform treatment strategies could be applied to a large fraction of TNBC patients.

* Finally, the authors describe mitochondrial cell death priming mechanism to explain the differences observed in drug sensitivity during co-culture. Though the depth of this mechanistic insight is limited the results do portend a general approach that could be used to sensitize TNBC to current treatments. Namely, BH3 mimetics are clinically relevant drugs and the connection between fibroblast effects and mitochondrial priming suggest that alternative methods to change mitochondrial priming could be attained by modulating the fibroblast. Ostensibly this indirect approach could alleviate some untoward side effects of BH3 mimetics.

We thank this reviewer for these very positive comments and for highlighting several high impact findings that emerge from our study. This last point highlighted by the reviewer regarding BH3 mimetics was very interesting to us, and highlighted an opportunity to expand upon our finding that fibroblasts appear to modulate drug responses by altering apoptotic priming. As the reviewer highlights, BH3 mimetic drugs are clinically relevant, but also not in common use for solid tumors, including breast cancers. Thus, an additional way to further test/validate our findings could be to determine if fibroblasts alter sensitivity to BH3 mimetic compounds. These drugs essentially modulate the apoptotic threshold by inhibiting anti-apoptotic signals, but they do not create pro-apoptotic signals, so we tested an "all-by-all" combinatorial matrix of a potent pro-apoptotic drug (teniposide) and a BH3 mimetic compound (ABT737), in the presence or absence of various fibroblast cells. Based on our understanding of the mechanism by which fibroblasts alter drug sensitivity of cancer cells, we predicted that the benefit of ABT737 would be very dependent on the growth environment. This is a non-trivial hypothesis, as currently BH3 mimetic compounds are being used based on the tumor subtype/genotype without consideration of the growth environment. Our new experiments and analyses that address this hypothesis are shown in a new Figure 6. Overall, we found that ABT737 did not kill BT20 cells when given as a single agent, but was synergistic with the apoptotic activator teniposide. The combination drug synergy was further enhanced when these drugs were applied to BT20 cells grown in a drug sensitizing environment (C12385). Additionally, the drug synergy was completely blocked when these drug combinations were tested in a potent drug de-sensitizing environment (HADF). Predicting combinatorial drug interactions has always been challenging, so our mechanistic insights, which helped here to predict a drug-drug-environment interaction should have real value to the community. Again, we thank the reviewer for this excellent suggestion.

Overall impression: This manuscript is very well-written, easy to interpret and takes on an important problem in cancer treatment using a systems based approach. Several novel-insights were made and innovative new approaches were developed and implemented with a high degree of experimental rigor.

Again, we thank the reviewer for this kind appraisal of our work.

*The study does lack significant mechanistic insights. The idea that mitochondrial priming is altered is interesting but how that occurs in one co-culture versus another is not explored. Clearly these authors are well-positioned to identify those factors that drive these differences. The question is whether that should be explored in more detail here. Clearly, identifying the precise effector molecules would raise the impact of this manuscript, but it would also likely make it more appealing to journal with a larger readership.

We absolutely agree with the reviewer that understanding how fibroblasts alter mitochondrial priming is a critically essential next step. Additionally, as highlighted by the reviewer, we do not view this as being a question that can be easily answered, and doing so will likely extend beyond the scope of this current study. Priming was first characterized by Tony Letai's group more than a decade ago, and currently, there are very few mechanistic insights about how cells regulate priming. It is clear that priming cannot be predicted by mRNA or protein expression data, likely due to the requirement for correct stoichiometry of many pro- and anti-apoptotic proteins as well as a strong spatial regulatory component. These studies will be the focus of our future efforts, and we have edited some portions of the discussion to highlight these challenges.

However, perhaps some further demonstration of how a key finding within the authors' co-culture dataset could also show increased efficacy in an in vivo model would be appropriate and still be within the scope of this publication.

The reviewer highlights another important consideration, that our study does not currently extend these findings to an in vivo model. However, we think it is worth highlighting the entirely non-trivial nature of these experiments for addressing our finding that fibroblasts from different tissues modulate drug responses in tissue-specific ways. Genetic systems are likely not feasible, as fibroblasts can be generated by a number of different precursor cells (and how these cells are created is still a matter of debate). Thus, to test these phenotypes in an in vivo model, we would co-inject tumor cells and primary fibroblast of different origin into the hind flank, or more likely, the mammary fat pad. Primary non-transformed fibroblast cells will not seed/grow well in this environment. Thus, in prior studies where this type of experiment has been attempted cells were injected with matrigel to support colonization and growth (see for example Marusyk, et al. Cancer Research 2016; Elenbaas et al. Genes and Development 2001). We have added a new Supplemental Figure 11 which shows that matrigel will mask the phenotypes induced by drug sensitizing and drug de-sensitizing fibroblasts. Additionally, in the study from the Polyak group (Marusyk et al.) the authors note that many/most of the fibroblasts that existed within the xenograft tumors appear to be of mouse origin, rather than the human fibroblasts that were injected. For these reasons, we think that a xenograft co-culture in vivo model may not be interpretable, as this would not really be testing the phenotypes that we found. To address this issue, we are currently exploring two separate avenues, which are now highlighted in the discussion as important future directions: 1) we hope to learn the molecular mechanisms by which fibroblasts alter the apoptotic priming state of cancer cells. Once this information is learned, mimicking or blocking these molecular interactions in an autochthonous genetic model system would be a valuable way to test our findings in vivo. 2) We and our collaborators are developing synthetic bio-inert polymers to support the growth of fibroblasts and other cell types in vivo without use of matrigel. Both of these are long-term (likely multi-year) efforts, which we do not think will fit within the scope of the current manuscript. We hope that our addition of several pieces of new data, combined with the reviewer's overall positive impression of our work, have sufficiently strengthened this study for publication.

Thank you for sending us your revised manuscript, related to your previous submission MSB-18-8322. We have now heard back from reviewer #2 who was asked to evaluate your manuscript. As you will see below, reviewer #2 is satisfied with the performed revisions and is supportive of publication.

Before we formally accept the study for publication, we would ask you to address some remaining editorial issues listed below.

REFEREE REPORTS.

Reviewer #2:

In original manuscript, I had a very favorable overall review, however, I made two suggestions that I thought would increase the impact of this work. The first was expand on the concept that altering mitochondrial priming through the use of BH3 mimetics to better codify the mechanism proposed by the authors that fibroblasts from distinct origins differentially alter drug sensitivity. The authors have now capitalized on that suggestion and added new data to support this notion. I believe that this has added much needed mechanistic support to their original hypothesis. Second, I had suggested that in vivo confirmation of their in vitro co-culture findings would be a helpful to validate differential sensitivities observed with distinct fibroblast co-cultures. The authors explain in their rebuttal that this is not a trivial experiment due the technical limitation that transplanted human fibroblasts injected as a mixture with cancer cells in murine xenograft models leads to replacement of the human fibroblast population by fibroblasts of mouse origin. While this explanation is sound, I am still left wanting for these data. The authors do address this deficiency in the discussion section of the manuscript and experimentally demonstrate some of the limitations described as supplemental data. Overall, my enthusiasm for the additional mechanistic insights provided regarding the mitochondrial priming model outweigh the negative impact associated with the lack of in vivo validation.

Corresponding Author Name: Michael J. Lee

Manuscript Number: MSB-18-8322R